# DeepITE: Designing Variational Graph Autoencoders for Intervention Target Estimation

**Hongyuan Tao**[*]
Ant Group
Hangzhou, China
thy.qy@antgroup.com

**Hang Yu**[*]
Ant Group
Hangzhou, China
hyu.hugo@antgroup.com

**Jianguo Li**[†]
Ant Group
Hangzhou, China
lijg.zero@antgroup.com

## Abstract

Intervention Target Estimation (ITE) is vital for both understanding and decision-making in complex systems, yet it remains underexplored. Current ITE methods are hampered by their inability to learn from distinct intervention instances collaboratively and to incorporate rich insights from labeled data, which leads to inefficiencies such as the need for re-estimation of intervention targets with minor data changes or alterations in causal graphs. In this paper, we propose DeepITE, an innovative deep learning framework designed around a variational graph autoencoder. DeepITE can concurrently learn from both unlabeled and labeled data with different intervention targets and causal graphs, harnessing correlated information in a self or semi-supervised manner. The model's inference capabilities allow for the immediate identification of intervention targets on unseen samples and novel causal graphs, circumventing the need for retraining. Our extensive testing confirms that DeepITE not only surpasses 13 baseline methods in the Recall@k metric but also demonstrates expeditious inference times, particularly on large graphs. Moreover, incorporating a modest fraction of labeled data (5-10%) substantially enhances DeepITE's performance, further solidifying its practical applicability. Our source code is available at https://github.com/alipay/DeepITE.

## 1 Introduction

Causal analysis in complex systems encompasses a series of steps beginning with causal discovery [1], which aims to delineate the causal structure, followed by the identification and estimation of causal effects [2]. Within this framework, a critical yet often overlooked component is Intervention Target Estimation (ITE) [3], alternatively known as Intervention Recognition [4]. ITE is the process of pinpointing which variables in a system have been subject to intervention, particularly when such interventions are opaque or not directly manipulable. This process not only fosters a deeper understanding of the causal mechanisms driving specific outcomes, which resonates with the principles of explainable AI (XAI), but also plays a pivotal role in recognizing variables that can be strategically altered to produce desired effects, aligning with the concept of algorithmic recourse [5].

To illustrate, consider the application of ITE in root cause analysis (RCA) within a microservices system. These systems consist of a network of services working in concert to deliver software functionality. When a service failure occurs, such as a system outage or performance degradation, it becomes imperative to identify the root cause. ITE is the key that unlocks definitive insight into the RCA process. It methodically pinpoints the specific services whose malfunctions—stemming from network issues, hardware failures, or security breaches—lead to the anomalies in question. ITE not

---

[*]Equal contribution.
[†]Corresponding author.

38th Conference on Neural Information Processing Systems (NeurIPS 2024).

only equips operators with a clear and logical explanation for the system's alerts, enhancing their comprehension of the issues at hand, but it also empowers them to implement immediate and effective countermeasures and fortify the system against future incidents. Beyond RCA in microservice systems, ITE's applicability extends to a multitude of domains, from unraveling the genetic factors involved in diseases within biomedicine, to tracing the determinants of user behavior for marketing.

Unfortunately, the field of ITE has not been thoroughly explored, often resulting in intervention targets being relegated to secondary outputs from causal discovery rather than being a dedicated field of inquiry. Only recently, Varici *et al.* [3, 6] pioneered the exclusive study of learning intervention targets in linear SCMs. In parallel, Li *et al.* [4] approached the problem from the perspective of RCA, coining it intervention recognition. Finally, Yang *et al.* [7] extended ITE to non-linear SCMs. This handful of methods can only identify the targets for an intervention instance[3] with a large sample size, relying on an accompanying dataset of known observational data, all within the confines of a fixed causal graph. The drawbacks of these strategies are twofold: **From the learning perspective**, they independently map data to intervention targets for each intervention instance, disregarding the potential correlations among distinct instances. In RCA scenarios, for example, various incidents could stem from the same underlying service problem. A collaborative learning approach, which considers all instances collectively, could more effectively elucidate the data-intervention target relationships. Moreover, these methods often neglect the labeled data that are often available, such as those obtained from controlled chaos engineering exercises that identify specific services as failure root causes. Consequently, opportunities to refine and expedite future similar analyses are lost. **From the inference standpoint**, slight changes in the data or in the graph structure necessitate a burdensome and complete re-estimation of intervention targets. In RCA contexts, this means that despite shared causality between distinct incidents, we still require piecemeal analyses for each new occurrence, leading to extended system downtime and delayed resolutions. Additionally, the premise that a large volume of data pertains to a uniform set of intervention targets is restrictive. Such data are challenging to obtain, which further complicates the assurance of these methods' performance.

Addressing these shortcomings, we introduce DeepITE, an innovative deep-learning solution that disentangles the learning and inference processes. In particular, **we design a variational graph autoencoder (VGAE) that can concurrently learn across diverse causal graphs and sets of intervention targets in a self-supervised or semi-supervised mode, thus, effectively harnesses correlated information to unravel the intricate relationship between input data and intervention targets. Once the VGAE is trained, its inference model can instantly identify intervention targets for new, unseen samples with different interventions and causal graphs, all without the need to retrain or refer to observational data.** Specifically, leveraging the principle that interventions entail the removal of all incoming edges to intervention targets, our VGAE framework is designed to estimate the probability of edge removal for each node, thereby identifying the intervention targets. The generative model within the VGAE employs a non-linear Graph Neural Network (GNN), an extension of linear SCMs that adheres to causal factorization and meets the criteria for causal interventions, across causal graphs of various structures and sizes. This theoretical foundation ensures robust ITE capabilities. The generative model accepts exogenous noise variables $\mathbf{u}$, the adjacency matrix of the observational causal graph $\mathbf{A}$, a Bernoulli-distributed intervention indicator $\boldsymbol{\gamma}$ characterizing edge removal probability, and outputs the distribution of endogenous variables $\mathbf{x}$. Conversely, the inference model interprets a given sample of endogenous variables $\mathbf{x}$ to infer distributions of exogenous variables $\mathbf{u}$ and the intervention indicator $\boldsymbol{\gamma}$, all of which rely on the causal graph $\mathbf{A}$. We employ graph attention networks (GAT) as the backbone due to their flexibility and scalability. This VGAE—comprising both the generative and inference models—can be trained in a self-supervised manner by maximizing the evidence lower bound (ELBO) or can leverage labeled intervention targets when available for semi-supervised learning.

Our contributions can be summarized as follows:

- We propose a novel VGAE architecture tailored for ITE, termed DeepITE. It excels in collaborative learning from varying causal structures and interventions, negating the need for retraining with each new instance.
- We establish self-supervised and semi-supervised training approaches for DeepITE, allowing it to autonomously discern intervention targets and enhance accuracy through the integration of labeled data from controlled experiments.

---

[3]We define an intervention instance as one manipulation on the causal system. We can then collect a set of data with the same intervention targets for this instance.

- Extensive experiments show that DeepITE surpasses 13 baseline methods by a large margin on average in terms of *Recall@k* with competitive inference time, especially for large graphs.

## 2   Related Works

In this section, we briefly review the literature on ITE. Moreover, we notice that the realm of ITE is interconnected with causal explanations and RCA. These three concepts demonstrate a considerable degree of overlap (cf. [8, 4, 9]), suggesting that methods from each domain can not only inform and enhance one another but also be utilized in a complementary fashion. We therefore refer the readers to Appendix B for further discussion on causal explanations and RCA.

The limited literature on ITE approaches bifurcates, with one camp focusing on incidental estimation of intervention targets via causal discovery and the other dedicated solely to identifying intervention targets within a given causal framework. The former includes methods such as UT-IGSP [10], which seeks to recover an interventional Markov equivalence class (I-MEC) through permutation searches but is hampered by sample inefficiency and limited scalability. Ghassami *et al.* [11] explore linear structural causal models (SCMs) yet may struggle with complexity in diverse data settings. For causal insufficient systems, Jaber *et al.* [12] propose $\Psi$-FCI for matching interventional distributions to causal graphs and intervention target pairs, contending with exponential growth in complexity as the number of variables increases. Mooij *et al.* [13] alternatively propose a method leveraging context variables for integrating interventional datasets, but this method suffers from scalability issues with large graphs. To overcome this problem, RCD [9] further adapts the $\Psi$-FCI algorithm in [12] to the $\Psi$-PC algorithm to expedite the process in causally sufficient systems. The second approach, exemplified by CITE [3] and PreDITEr [6], zeros in on ITE by contrasting precision matrices from observational and interventional data, achieving scalability at the cost of being initially restricted to linear Gaussian SCMs. LIT [7] explores non-linear SCMs through non-linear ICA, still carrying quadratic complexity. Alternatively, CI-RCA [4] conducts ITE by detecting shifts in probability distributions of a variable conditioned on a variable's parents via hypothesis testing. Both groups of methods share critical disadvantages: they are vulnerable to even minor changes in data or causal graphs during both learning and inference, and they underutilize labeled data from controlled experiments.

## 3   Preliminaries

This section lays the groundwork for our study by introducing SCMs and Pearl's Causal Hierarchy.

**Structural Causal Models (SCMs)**: Given a set of variables $\mathbf{x} = [x_1, \ldots, x_d]$, SCMs present a formal mechanism to represent causal relations among them. An SCM is composed of two primary components: a set of structural equations and a causal graph. The structural equations take the form:

$$x_i := f_i(\mathrm{Pa}(x_i), u_i), \quad i = 1, 2, \ldots, m, \tag{1}$$

where $f_i$ is a deterministic function, $\mathrm{Pa}(x_i)$ denotes parent variables that exert direct causal influence on $x_i$, and $u_i$ signifies unobserved exogenous variables capturing influences not represented by other variables in $\mathbf{x}$. We typically invoke causal sufficiency, assuming the $u_i$ are jointly independent, thereby ruling out hidden confounders. The use of the assignment symbol ":=" instead of an equality sign underscores the asymmetry of the causal relationship. The corresponding causal graph $\mathcal{G}$ (see Figure 1(a) for an example) induced by the SCM is typically a directed acyclic graph (DAG) with vertex set $\mathbf{x} \cup \mathbf{u}$ and directed edges from each variable on the right hand side (RHS) of a structural equation (1) to the variable on the left hand side, thus delineating the causal dependencies.

**Pearl's Causal Hierarchy (PCH)**: Under the SCMs framework, PCH categorizes causal inference into a three-tiered structure reflective of the cognitive processes of "seeing" (observational), "doing" (interventional), and "imagining" (counterfactual). The initial tier addresses observational queries using SCMs as conventional probabilistic models to describe statistical associations.

Progressing to the second tier of PCH, SCMs distinguish themselves from standard probabilistic models by enabling the assessment of outcomes resulting from active interventions or manipulations, captured by the notions of *do-operator* and *graph surgery*. Here, the do-operator $\mathrm{do}(x_i = x_i)$ represents an intervention that sets the variable $x_i$ to value $x_i$, while graph surgery alters the corresponding causal graph by removing all incoming edges to the intervened-upon variable $x_i$. Interventional queries are then addressed by performing probabilistic inference in the modified graph, which often reveals new conditional independencies due to the excision of edges. For example, the interventional

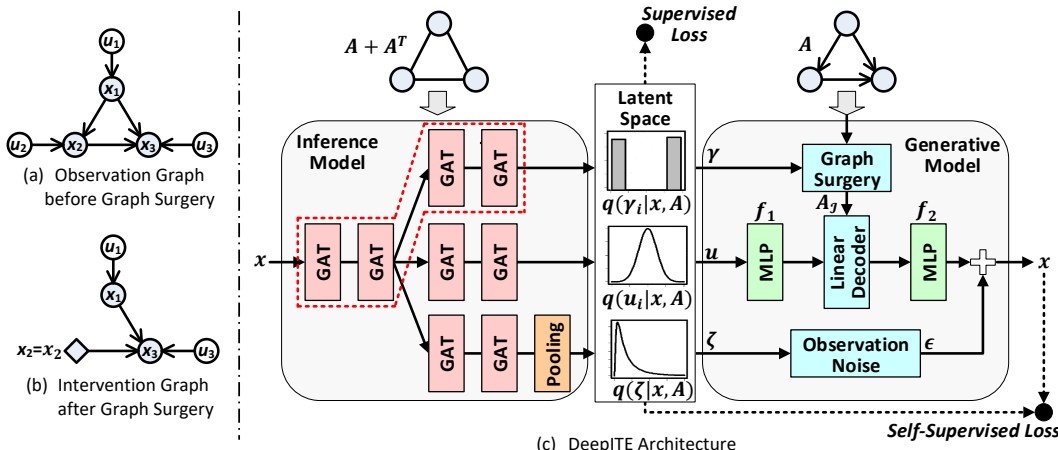

Figure 1: **Left Panel**: Illustration of the do operator and the corresponding graph surgery: (a) The observation graph $\mathcal{G}$; (b) The intervention graph $\mathcal{G}_{\mathcal{I}}$ for $\mathrm{do}(\mathrm{x}_2 = x_2)$. **Right Panel** (c): The DeepITE architecture has an inference and a generative model. The inference model uses a three-branch GAT to link endogenous variables $\mathbf{x}$ to posterior distributions of intervention indicators $\gamma_i$, exogenous variables $\mathrm{u}_i$, and observation noise precision $\zeta$. The generative model then synthesizes $\mathbf{x}$ given these latent variables following Eq. (7) plus observation noise $\epsilon$.

distribution $p(\mathrm{x}_3 | \mathrm{do}(\mathrm{x}_2 = x_2))$ for the SCM in Figure 1(b) is obtained via probabilistic inference with regard to (w.r.t.) the intervention graph:

$$p(\mathrm{x}_3 | \mathrm{do}(\mathrm{x}_2 = x_2)) = \sum_{\mathrm{x}_1} p(\mathrm{x}_1) p(\mathrm{x}_3 | \mathrm{x}_1, \mathrm{x}_2 = x_2), \tag{2}$$

contrasting with the conditional distribution in the original graph (i.e., Figure 1(a)):

$$p(\mathrm{x}_3 | \mathrm{x}_2 = x_2) = \sum_{\mathrm{x}_1} p(\mathrm{x}_1 | \mathrm{x}_2 = x_2) p(\mathrm{x}_3 | \mathrm{x}_1, \mathrm{x}_2 = x_2). \tag{3}$$

The key distinction is the marginal $p(\mathrm{x}_1)$ in (2) versus the conditional $p(\mathrm{x}_1 | \mathrm{x}_2 = x_2)$ in (3), reflecting that the causal relationship between $\mathrm{x}_1$ and $\mathrm{x}_2$ is broken by the intervention $\mathrm{do}(\mathrm{x}_2 = x_2)$.

Finally, there are *counterfactual* queries about what would or could have been, given that something else was in fact observed. We refer the readers to [14] as this topic is beyond the scope of our paper.

## 4   Problem Formulation

As discussed in the introduction, intervention target estimation (a.k.a. intervention recognition) is a pivotal component within the landscape of causal analysis, addressing the question of which nodes within a given causal system should be subjected to intervention in order to best explain the given interventional data. This inquiry draws upon the framework established by SCMs and operationalizes the principles enshrined in the second tier of Pearl's Causal Hierarchy. More formally,

**Definition 1.** *Given a causal graph $\mathcal{G}$ with variables $\mathbf{x}$, the observational data, and the interventional data corresponding to a certain intervention on a subset of variables $\mathbf{x}_{\mathcal{I}} \subseteq \mathbf{x}$, the task of intervention target estimation is to identify $\mathbf{x}_{\mathcal{I}}$.*

Here, we maintain the assumptions of causal sufficiency and the acyclicity of the causal graph. Note that while the DAG structure (i.e., the adjacency matrix $A$) is assumed to be known, the explicit forms of the structural equations remain unspecified. In comparison with the interventional queries mentioned in Section 3, which presuppose knowledge of where interventions have occurred and seek to determine their effects, ITE is the inverse process: it starts with the consequences of interventions and works backward to identify the sources of these perturbations. Essentially, we are solving for the *origin* of the observed interventional data, rather than predicting their *impact*.

In this paper, we innovatively solve the problem of intervention target estimation from the perspective of graph surgery. Recognizing that the interventional data align best with the intervention graph (see Figure 1(b)), our objective is to discover the subset of nodes $\mathbf{x}_{\mathcal{I}}$ such that, upon hypothetically removing the incoming edges to these nodes, the modified interventional model most accurately reflects the presented interventional data. Furthermore, We aim to create a singular model capable of pinpointing distinct sets of intervention targets for individual samples, each associated with its unique causal graph, while eliminating the need for observational data during inference. This represents

a significant shift from conventional ITE methods that independently identify shared intervention targets for each intervention instance based on both observational and interventional data within the context of a fixed causal graph. Additionally, we seek to enhance model performance by incorporating labeling information during the training phase, which is a step forward in refining ITE processes.

## 5  DeepITE

To move forward to the above objectives, we present DeepITE, a VGAE that can estimate the probability of edge presence within the latent space. The architecture of DeepITE is showcased in Figure 1(c). We will first introduce the generative and inference models within the VGAE. Subsequently, we will explicate the self and semi-supervised methods to train the inference model.

### 5.1  Generative Model

The chief aim of the generative model is to recreate the observed variables $\mathbf{x}$ given the exogenous noise variables $\mathbf{u}$, the adjacency matrix $\boldsymbol{A}$ of the causal DAG, and a set of nodes $\mathcal{I}$ that have been intervened upon. Notably, when $\mathcal{I}$ is an empty set, the model is capable of recovering the observational distribution.

Specifically, when the structural equations are linear, they can be succinctly written as $\mathbf{x} = \boldsymbol{A}^T\mathbf{x} + \mathbf{u}$. Therefore, given $\mathbf{u}$ and $\boldsymbol{A}$, we can derive $\mathbf{x}$ through a linear decoder:
$$\mathbf{x} = (\boldsymbol{I} - \boldsymbol{A}^T)^{-1}\mathbf{u}. \tag{4}$$
Drawing inspiration from DAG-GNN [15], we extend this formulation to non-linear scenarios with:
$$\mathbf{x} = f_2\big((\boldsymbol{I} - \boldsymbol{A}^T)^{-1}f_1(\mathbf{u})\big), \tag{5}$$
where $f_1$ and $f_2$ are non-linear, component-wise learnable functions. In practice, these functions are executed by MLPs, which serve as universal approximators. Assuming $f_2$ is invertible, the aforementioned decoder corresponds to a conglomerate of non-linear structural equations [15]:
$$f_2^{-1}(\mathbf{x}) = \boldsymbol{A}^T f_2^{-1}(\mathbf{x}) + f_1(\mathbf{u}). \tag{6}$$
This setup implies that when $f_1$ and $f_2^{-1}$ are suitably expressive, they can transform $\mathbf{u}$ and $\mathbf{x}$ into a space where their causal interrelations are aptly described by linear structural equations. The decoder, as specified in Eq. (5), exhibits inductiveness, enabling generalization to new nodes, edges, or graph schemas by only modifying the adjacency matrix $\boldsymbol{A}$ while preserving the learned functions $f_1$ and $f_2$. This decoder displays a particular characteristic, ratified by the following proposition:

**Proposition 1.** *For a GNN layer as defined in Eq.* (5)*, and denoting* $\mathrm{An}(i)$ *as the ancestor nodes of node* $i$ *with the extension* $\mathrm{An}^*(i) = \mathrm{An}(i) \cup i$*, each output feature* $\mathrm{x}_i$ *exclusively acquires information from its own and all ancestor input features* $\mathbf{u}_{\mathrm{An}^*(i)}$*.*

*Proof.* See Appendix C. □

Owing to this property, this decoder (5) satisfies causal factorization and captures causal intervention, as proven below.

**Proposition 2.** *(causal factorization) The decoder defined in Eq.* (5) *conforms to causal factorization* $p(\mathrm{x}|\mathbf{u}, \boldsymbol{A}) = \prod_i p(\mathrm{x}_i|\mathbf{u}_{\mathrm{An}^*(i)})$*, that is, each endogenous variable* $\mathrm{x}_i$ *can be expressed as a function of its exogenous variable* $\mathrm{u}_i$ *and those of its causal ancestors.*

**Proposition 3.** *(causal intervention) The decoder defined in Eq.* (5) *captures causal interventions* $\mathrm{do}(\mathbf{x}_{\mathcal{I}} = x_{\mathcal{I}})$ *by replacing the original adjacency matrix* $\boldsymbol{A}$ *in Eq.* (5) *with the one corresponding to the post-intervention graph after graph surgery.*

*Proof.* See Appendices D and E. □

As established in Section 4, ITE resides within the second tier of PCH. Within this framework, any model purporting to tackle the ITE challenge must adeptly manage both observational and interventional data. The above propositions bridge this requirement, affirming DeepITE's competency in fulfilling the ITE task. Specifically, these propositions serve as the key to unlocking the model's ability to honor the causal structure inherent in the data and to emulate the effects of interventions.

In light of Proposition 3, to manipulate the intervened nodes $\mathcal{I}$ in the decoder (5), we introduce a Bernoulli distributed variable $\gamma_i$ for each node $\mathrm{x}_i$: $\gamma_i = 0$ means $\mathrm{x}_i$ is intervened, and thus, all incoming edges of $\mathrm{x}_i$ is removed during the graph surgery. The variable $\gamma_i$ is henceforth referred to as the intervention indicator. The corresponding intervened adjacency matrix is given by $\boldsymbol{A}_{\mathcal{I}} = (\gamma^T \mathbf{1}) \odot \boldsymbol{A}$, where $\mathbf{1}$ is a column vector of all ones and $\odot$ denotes Hadamard product. As a result, the decoder, inclusive of interventions, is delineated as:
$$\mathbf{x} = \mathrm{Dec}(\mathbf{u}, \gamma, \boldsymbol{A}) = f_2\Big(\big(\boldsymbol{I} - \boldsymbol{A}_{\mathcal{I}}^T\big)^{-1}f_1(\mathbf{u})\Big). \tag{7}$$

When $\gamma_i = 1$ for all $i$, there is no intervention (i.e., $\boldsymbol{A}_{\mathcal{I}} = \boldsymbol{A}$) and the above decoder can describe the observational distribution.

To facilitate the reparameterization trick in VAE, we assume the exogenous variables $\mathbf{u}$ are standard normal distributions: $\mathbf{u}_i \sim \mathcal{N}(0, 1)$. Finally, since we do not have access to the true structural equations, we introduce the observation noise $\epsilon \sim \mathcal{N}(0, \zeta^{-1})$ to (7), so as to account for the model uncertainty associated with the above decoder (7). Here, $\zeta$ denotes the inverse variance of the noise, and we impose a non-informative Jeffrey's prior on $\zeta$, that is, $p(\zeta) \propto 1/\zeta$.

Collectively, the overall generative model can be factorized as:

$$p(\mathbf{x}, \mathbf{u}, \boldsymbol{\gamma}, \zeta | \boldsymbol{A}) = p(\mathbf{x} | \mathbf{u}, \boldsymbol{\gamma}, \boldsymbol{A}, \zeta) p(\zeta) \prod_{i=1}^{m} p(\mathbf{u}_i) p(\gamma_i), \tag{8}$$

where

$$p(\mathbf{u}_i) = \mathcal{N}(0, 1), \quad p(\gamma_i) = \mathrm{Bern}(\pi) \quad \forall i,$$
$$p(\zeta) \propto 1/\zeta, \quad p(\mathbf{x} | \mathbf{u}, \boldsymbol{\gamma}, \boldsymbol{A}, \zeta) = \mathcal{N}\big( \mathrm{Dec}(\mathbf{u}, \boldsymbol{\gamma}, \boldsymbol{A}), \zeta^{-1} \boldsymbol{I} \big). \tag{9}$$

Here, $\pi$ denotes the probability of taking 1 in a Bernoulli distribution and $\boldsymbol{I}$ is the identity matrix.

**Discussion on Hard versus Soft Interventions**: Hard interventions, characterized by the removal of incoming edges as part of graph surgery, contrast with soft interventions, which modify the causal mechanism without complete elimination. For instance, for an intervened node $\mathbf{x}_i$, a soft intervention would replace the original structural equation $\mathbf{x}_i := f_i(\mathrm{Pa}(\mathbf{x}_i), \mathbf{u}_i)$ with an updated version $\mathbf{x}_i := f_i'(\mathrm{Pa}(\mathbf{x}_i), \mathbf{u}_i)$, where $f_i \neq f_i'$, thereby altering the generative process while maintaining the graph's structure. In our generative model, the intervention indicator $\gamma_i$ is a Bernoulli variable, allowing the learning of edge removal probability from data $\mathbf{x}$. A probability of $\gamma_i = 0$ being one indicates a hard intervention, whereas any other value suggests a soft intervention. This nuanced approach allows DeepITE to offer a spectrum between hard and soft interventions based on given data.

## 5.2 Inference Model

The pinnacle goal of the inference model within DeepITE is to determine the probability that a node $i$ has undergone an intervention based on the observed data, succinctly expressed as $p(\gamma_i = 1 | \mathbf{x})$. To achieve this, we aim to compute the exact posterior $p(\mathbf{u}, \boldsymbol{\gamma}, \zeta | \mathbf{x}, \boldsymbol{A})$. However, the intricate nature of this posterior necessitates approximation through a tractable inference model $q(\mathbf{u}, \boldsymbol{\gamma}, \zeta | \mathbf{x}, \boldsymbol{A})$ [16, 17]. Specifically, the inference model can be factorized in a manner akin to the generative model as:

$$q(\mathbf{u}, \boldsymbol{\gamma}, \zeta | \mathbf{x}, \boldsymbol{A}) = q(\zeta | \mathbf{x}, \boldsymbol{A}) \prod_{i=1}^{m} q(\mathbf{u}_i | \mathbf{x}, \boldsymbol{A}) q(\gamma_i | \mathbf{x}, \boldsymbol{A}). \tag{10}$$

The variational distributions on RHS are parametrized by:

$$q(\mathbf{u}_i | \mathbf{x}, \boldsymbol{A}) = \mathcal{N}\big( \mu_i(\mathbf{x}, \boldsymbol{A}), \sigma_i^2(\mathbf{x}, \boldsymbol{A}) \big), \tag{11}$$

$$q(\gamma_i | \mathbf{x}, \boldsymbol{A}) = \mathrm{Bern}\big( \omega_i(\mathbf{x}, \boldsymbol{A}) \big), \tag{12}$$

$$q(\zeta | \mathbf{x}, \boldsymbol{A}) = \mathrm{Lognormal}\big( \mu_\zeta(\mathbf{x}, \boldsymbol{A}), \sigma_\zeta^2(\mathbf{x}, \boldsymbol{A}) \big), \tag{13}$$

where the Bernoulli distribution (26) can be well approximated using the Gumbel-Softmax reparameterization trick [18, 19]. The parameters of the variational $q$ distributions in (25)-(27) are derived from a network based on Graph Attention Networks (GAT). While any inductive spatial GNN can be used as the inference network in DeepITE, we choose GAT since it provides the flexibility and scalability necessary for our model. This flexibility stems from GAT's ability to dynamically weigh the importance of different nodes, thus allowing the variational distribution given by the inference network better approximate the exact posterior distribution. This advantage is further demonstrated in Appendix G.6, where we replace the GAT encoder with the encoder of DAG-GNN.

The architecture features an initial dual-layer GAT for feature extraction, followed by three specialized branches dedicated to the parameters of $\mathbf{u}$, $\boldsymbol{\gamma}$, and $\zeta$. Note that the parameters of $\mathbf{u}$ and $\boldsymbol{\gamma}$ can be regarded as node-level features, while those of $\zeta$ as graph-level features. As such, the final $\zeta$ branch includes a pooling layer to yield the graph-level features. The complete inference network is depicted in Figure 1(c).

It is pertinent to mention that the graph associated with the GATs is undirected, in contrast to the directed nature of the causal graph. This design choice is motivated by the need for the variational update of a variable to account for all elements within its Markov blanket, which includes the parents,

children, and co-parents of the node [20]. An undirected graph facilitates the message passing process within this Markov blanket by removing the constraints imposed by edge directionality. Moreover, the use of GAT ensures that the resulting model is inductive, enabling its application to new nodes within the graph and even entirely new graph structures.

**Relation to DAG-GNN [15]**: DAG-GNN's objective is to infer the structure of a DAG (i.e., the zero pattern of $\boldsymbol{A}$) from the provided data, a process known as causal discovery. The architecture of DAG-GNN employs a generative model expressed as $\mathbf{x} = f_2((\boldsymbol{I} - \boldsymbol{A}^T)^{-1} f_1(\mathbf{u}))$ and an inference model as $\mathbf{u} = f_4((\boldsymbol{I} - \boldsymbol{A}^T) f_3(\mathbf{x}))$, both of which are differentiable with respect to $\boldsymbol{A}$. This differentiability is crucial as it enables the learning of $\boldsymbol{A}$ through gradient descent. On the other hand, DeepITE enhances the generative model by integrating an intervention indicator, which facilitates the adaptation of the model to account for interventions via graph surgery. Furthermore, DeepITE's inference model seeks to closely approximate the true posterior $p(\mathbf{u}, \boldsymbol{\gamma}, \zeta | \mathbf{x}, \boldsymbol{A})$. Unlike DAG-GNN, which may only collect messages from the parents of a node, DeepITE's model is designed to aggregate messages from all nodes within the Markov blanket of a given node $i$. This comprehensive approach ensures that DeepITE's inference model is not as restrictive as DAG-GNN's and is better suited for ITE tasks.

**Relation to VACA [21]**: VACA sets out to perform causal queries utilizing observational data within the framework of the VGAE. It hinges on Message Passing Neural Networks (MPNNs) for both the generative and inference models. A prerequisite for VACA to perform observational and interventional queries is that the generative model's number of MPNN layers must at least be $\delta - 1$ given that $\delta$ is the graph diameter.[4] This criterion ensures that the information propagated within the graph can reach from one end to the other, thereby reflecting the global structure necessary for accurate causal inference. DeepITE aligns with this requirement for effectively performing ITE. However, DeepITE distinguishes itself by employing the generative model, specified in Eq. (7). It keeps the number of GNN layers to be one regardless of graph diameter, while satisfying causal factorization and intervention conditions (cf. Proposition 2-3). DeepITE thereby overcomes the limitations imposed by VACA's dependence on graph diameter, offering a substantial benefit for collective learning on graphs with different sizes.

## 5.3  Self and Semi-Supervised Learning

DeepITE's learning strategies encompass self-supervised learning, which automates the identification of intervention targets from unlabeled data, and semi-supervised learning, which refines the model's performance by integrating labeled data. The training process is summarized in Algorithm 1.

**Self-Supervised Learning**: Given the generative and inference model, we can learn their parameters jointly by maximizing the evidence lower bound (ELBO) of the log-likelihood of the given data $\mathbf{x}$:

$$\mathcal{L} = \mathbb{E}_q[\log p(\mathbf{x}, \mathbf{u}, \boldsymbol{\gamma}, \zeta | \boldsymbol{A})] + \mathbb{H}_q \leq \log p(\mathbf{x} | \boldsymbol{A}), \tag{14}$$

where $\mathbb{E}_q$ denotes expectation over the $q$ distribution in (10) and $\mathbb{H}_q$ denotes the entropy of the $q$ distribution. The derivation of the ELBO can be found in Appendix F. Note that $\mathcal{L}$ can be maximized via stochastic gradient ascent after using the reparameterization trick for normal and Bernoulli distributions [22, 23, 19].

**Semi-Supervised Learning**: Information regarding intervention targets may often be available in practice. For instance, in the case of RCA in cloud-native systems, the ground truth of intervention targets can be derived from resolved incidents and chaos engineering exercises. This ground truth data can be utilized to train the inference network, enabling it to more accurately identify intervention targets. In particular, the term $q(\gamma_i | \mathbf{x}, \boldsymbol{A})$ is replaced by the ground truth $\gamma_i^*$ when computing the ELBO $\mathcal{L}$, and an additional term is introduced to maximize the log-likelihood $\log q(\gamma_i^* | \mathbf{x}, \boldsymbol{A})$. By taking advantage of both labeled and unlabeled data, we can effectively train the inference model.

Once trained, the inference model of DeepITE becomes equipped to evaluate individual new samples against different causal graphs, directly deducing the intervention targets and thus circumventing the necessity of retraining for each new scenario. It can also distinguish between observational and interventional data directly as the former equates to the absence of intervention targets. Another significant feature of DeepITE is its independence from observational data during the testing phase; it relies solely on the interventional data input into the inference network. This is a distinct advantage over existing ITE methods, which consistently require observational data for ITE. In practice, we are primarily concerned with the intervention indicators $\boldsymbol{\gamma}$. Hence, during inference, we only need

---

[4]The diameter of a graph is the length of the shortest path between the most distanced endogenous nodes.

Table 1: Recall@k of different algorithms for detecting the intervened nodes from the Synthetic dataset. Graph-$m$ means DAGs with $m$ nodes.

| DATASET | Graph-50 | | Graph-100 | | Graph-500 | |
|---|---|---|---|---|---|---|
| METRICS | Recall@1 | Recall@5 | Recall@1 | Recall@5 | Recall@1 | Recall@5 |
| UT-IGSP | $0.224 \pm 0.015$ | $0.318 \pm 0.019$ | $0.079 \pm 0.007$ | $0.185 \pm 0.010$ | $0.016* \pm 0.002$ | $0.020* \pm 0.004$ |
| CITE | $0.098 \pm 0.009$ | $0.124 \pm 0.013$ | $0.044 \pm 0.003$ | $0.063 \pm 0.006$ | $0.007 \pm 0.001$ | $0.008 \pm 0.001$ |
| PreDITEr | $0.104 \pm 0.009$ | $0.122 \pm 0.012$ | $0.049 \pm 0.004$ | $0.066 \pm 0.006$ | $0.008 \pm 0.001$ | $0.008 \pm 0.001$ |
| TreeExplainer | $0.381 \pm 0.022$ | $0.510 \pm 0.017$ | $0.298 \pm 0.016$ | $0.448 \pm 0.009$ | $0.102 \pm 0.008$ | $0.152 \pm 0.002$ |
| ASV | $0.296 \pm 0.021$ | $0.390 \pm 0.022$ | $0.261 \pm 0.014$ | $0.323 \pm 0.017$ | $0.081 \pm 0.003$ | $0.140 \pm 0.005$ |
| ShapleyFlow | $0.552 \pm 0.017$ | $0.690 \pm 0.009$ | $0.378 \pm 0.009$ | $0.485 \pm 0.007$ | $0.124 \pm 0.005$ | $0.148 \pm 0.002$ |
| PWSHAP | $0.468 \pm 0.014$ | $0.610 \pm 0.012$ | $0.339 \pm 0.009$ | $0.454 \pm 0.008$ | $0.117 \pm 0.003$ | $0.195 \pm 0.003$ |
| CauseInfer | $0.561 \pm 0.002$ | $0.765 \pm 0.003$ | $0.554 \pm 0.002$ | $0.786 \pm 0.02$ | $0.559 \pm 0.003$ | $0.769 \pm 0.004$ |
| MicroHECL | $0.462 \pm 0.010$ | $0.587 \pm 0.009$ | $0.341 \pm 0.004$ | $0.400 \pm 0.004$ | $0.199 \pm 0.003$ | $0.241 \pm 0.004$ |
| MicroRCA | $0.647 \pm 0.004$ | $0.899 \pm 0.003$ | $0.623 \pm 0.004$ | $0.875 \pm 0.004$ | $0.436 \pm 0.003$ | $0.676 \pm 0.003$ |
| CausalRCA | $0.633 \pm 0.004$ | $0.894 \pm 0.004$ | $0.622 \pm 0.004$ | $0.863 \pm 0.004$ | $0.418 \pm 0.004$ | $0.630 \pm 0.004$ |
| CI-RCA | $0.615 \pm 0.002$ | $0.952 \pm 0.001$ | $0.631 \pm 0.002$ | $0.930 \pm 0.003$ | $0.623 \pm 0.004$ | $0.823 \pm 0.003$ |
| RCD | $0.495 \pm 0.003$ | $0.706 \pm 0.004$ | $0.440 \pm 0.004$ | $0.521 \pm 0.005$ | $0.325 \pm 0.002$ | $0.364 \pm 0.003$ |
| DeepITE (sep) | $\mathbf{0.723} \pm 0.002$ | $\mathbf{0.972} \pm 0.003$ | $0.685 \pm 0.004$ | $\mathbf{0.968} \pm 0.002$ | $\mathbf{0.642} \pm 0.003$ | $\mathbf{0.891} \pm 0.004$ |
| DeepITE (mix) | $0.718 \pm 0.003$ | $0.945 \pm 0.005$ | $\mathbf{0.690} \pm 0.004$ | $0.923 \pm 0.003$ | $0.627 \pm 0.003$ | $0.875 \pm 0.004$ |

to process $\mathbf{x}$ through the relevant branch of $\mathbf{x}$, as highlighted by the red dashed box in Figure 1(c), disregarding the other branches to optimize inference efficiency.

# 6    Experimental Results

In this section, we demonstrate the usefulness of DeepITE on three datasets, comprising one synthetically generated dataset, which provides a controlled environment to test the robustness and scalability of the framework, and two real-world datasets that introduce the complexity of genuine causal systems. We position DeepITE against 13 state-of-the-art (SOTA) methods, spanning three areas of relevance: Intervention Target Estimation (ITE), Explainable AI (XAI), and Root Cause Analysis (RCA), due to their intertwined nature (see more discussions in Section 2 and Appendix B).

- **ITE**: We select 3 methods: UT-IGSP [10], which learns intervention targets as a byproduct of causal discovery; CITE [3] and PreDITEr [6], both of which are dedicated to ITE.

- **XAI**: We opt for TreeExplainer [24], ASV [25], ShapleyFlow [26], and PWSHAP [27], 4 methods based on Shapley values. TreeExplainer only considers associations, whereas ASV, ShapleyFlow, and PWSHAP incorporate causation, accounting for the DAG structure.

- **RCA**: We pick 6 methods: CauseInfer [28], MicroHECL [29], MicroRCA [30], CausalRCA [31], CI-RCA [4], and RCD [9]. The last two aim to find intervention targets in a causal graph.

To facilitate a fair comparison, all methods are provided with the same ground truth causal graph, eschewing the need for graph construction from data for some RCA methods. More implementation details can be found in Appendix G.1. The performance is quantified using the *Recall@k* metric. *Recall@k* measures the proportion of true intervention targets (ITs) that are successfully captured within the top $k$ ranked candidates proposed by each method. This metric is widely adopted in the literature [4, 9, 29]. When $k = 1$, our goal is to pinpoint the intervention targets based on the highest-ranked candidate. We prioritize *Recall@k* because, in practice, false positives can be eliminated through further analysis, while false negatives are irrecoverable as they get lost among the numerous true negatives. All experiments report average results over 10 trials, with error bars representing a standard deviation ($\pm 1\sigma$) from the mean.

**Synthetic Data**: The synthetic data is generated following the method outlined in CI-RCA [4] and in Appendix G.2. We assess causal graphs with nodes ranging from 50 to 500 and corresponding edges from 100 to 5000, exhibiting different levels of complexity. In particular, DeepITE is evaluated in two configurations: DeepITE (sep), where separate models are trained for each graph size, and DeepITE (mix), where a single model is trained across all graph sizes and structures.

Analysis of the results in Table 1 reveals 5 key insights: (i) **Traditional ITE Methods Fall Short**: Such methods underperform with larger graphs with fewer samples due to their design for small, dense datasets (typically involving tens of nodes but with tens of thousands of samples) and lack of cross-instance learning, treating each intervention instance in isolation. As shown in Appendix G.3, they perform much better for small graphs with large sample size. (ii) **XAI Methods Face Challenges**: TreeExplainer lacks consideration for causal relationships in graphs. ASV and Shapley Flow, although

Table 2: Results of the Protein Signaling Data and the ICASSP-SPGC Data.

| DATASET | Protein Signaling | ICASSP-SPGC 2022 | | | |
|---|---|---|---|---|---|
| METRICS | Recall@1 | Recall@1 | Recall@5 | Root.Acc | Score |
| UT-IGSP | $0.579 \pm 0.018$ | - | - | - | - |
| CITE | $0.588 \pm 0.011$ | - | - | - | - |
| PreDITEr | $0.586 \pm 0.010$ | - | - | - | - |
| TreeExplainer | $0.434 \pm 0.020$ | $0.367 \pm 0.013$ | $0.687 \pm 0.010$ | $0.7401 \pm 0.0153$ | $0.3534 \pm 0.0298$ |
| ASV | $0.441 \pm 0.017$ | $0.449 \pm 0.010$ | $0.720 \pm 0.008$ | $0.7933 \pm 0.0117$ | $0.3820 \pm 0.0230$ |
| ShapleyFlow | $0.615 \pm 0.021$ | $0.677 \pm 0.013$ | $0.825 \pm 0.015$ | $0.9176 \pm 0.0131$ | $0.5312 \pm 0.0252$ |
| PWSHAP | $0.603 \pm 0.016$ | $0.488 \pm 0.009$ | $0.741 \pm 0.010$ | $0.8551 \pm 0.0109$ | $0.4233 \pm 0.0211$ |
| CauseInfer | $0.076 \pm 0.002$ | $0.278 \pm 0.003$ | $0.490 \pm 0.001$ | $0.5808 \pm 0.0084$ | $0.1139 \pm 0.0180$ |
| MicroHECL | $0.081 \pm 0.004$ | $0.323 \pm 0.010$ | $0.661 \pm 0.011$ | $0.7339 \pm 0.0134$ | $0.3697 \pm 0.0283$ |
| MicroRCA | $0.127 \pm 0.003$ | $0.246 \pm 0.004$ | $0.463 \pm 0.002$ | $0.5662 \pm 0.0089$ | $0.0721 \pm 0.0204$ |
| CausalRCA | $0.113 \pm 0.001$ | $0.308 \pm 0.004$ | $0.447 \pm 0.003$ | $0.5353 \pm 0.0078$ | $0.0617 \pm 0.0161$ |
| CI-RCA | $0.090 \pm 0.001$ | $0.559 \pm 0.002$ | $0.828 \pm 0.001$ | $0.9284 \pm 0.0055$ | $0.5650 \pm 0.0105$ |
| RCD | $0.214 \pm 0.002$ | $0.481 \pm 0.008$ | $0.757 \pm 0.005$ | $0.8768 \pm 0.0112$ | $0.4542 \pm 0.0216$ |
| DeepITE | $\mathbf{0.652} \pm 0.002$ | $\mathbf{0.881} \pm 0.002$ | $\mathbf{0.984} \pm 0.000$ | $\mathbf{0.9794} \pm 0.0023$ | $\mathbf{0.9085} \pm 0.0040$ |

aware of the causal structure, struggle with scalability similar to traditional ITE methods, as these methods demonstrate optimal performance on comparatively smaller graphs (around 10-20 nodes) (cf. [25, 26]). (iii) **RCA Methods Show Promise but Have Limitations**: RCA methods generally perform better than ITE and XAI approaches, with CI-RCA aligning exactly with the ITE task. However, these methods also face challenges in integrating multiple intervention instances effectively. (iv) **DeepITE Models Excel**: Both DeepITE (sep) and DeepITE (mix) outperform all benchmarked methods on Recall@1 and Recall@5 metrics, attributing success to a flexible model that fosters collaborative instance learning, independent of graph characteristics, enabling precise alignment of data with intervention targets. (v) **Competitive Performance within DeepITE Models**: The two DeepITE models (sep&mix) demonstrate competitive results, indicating that the model's inductive strength and its adaptability to various graph structures and sizes.

**Protein Signaling Dataset**: The description of the dataset is presented in Appendix G.4. The results, shown in the second column of Table 2, focus on the Recall@1 metric due to the graph's limited size. It is evident that DeepITE outshines competing methods, attributable to its capacity for collaborative learning across the entire dataset and its inherent adaptability. Traditional ITE and XAI methods trail closely behind, with their methodologies being particularly suited to smaller graphs that nevertheless have a substantial number of samples. In contrast, RCA methods exhibit weaker performance as techniques such as PageRank, DFS, BFS, or rank walk struggle to distinguish between nodes in such a compact network. Unlike these approaches, DeepITE demonstrates versatility in handling both small and large graphs, affirming its utility across a wide range of practical scenarios.

**ICASSP-SPGC 2022**: The details of this dataset can be found in Appendix G.4. Note that the absence of purely observational data in this dataset precludes the application of the ITE methods including UT-IGSP, CITE, and PreDITEr. For our evaluation, we continue to employ Recall@1 and Recall@5 metrics and additionally consider accuracy and the root-cause score, the latter two being an official recommendation. The root-cause score is calculated by subtracting the number of false positives from the number of true positives and then dividing by the total number of true intervention targets. The results of this multifaceted assessment are presented in the last four columns of Table 2. Once more, in comparison with the SOTA methods, DeepITE stands out by a large margin of above 20% in Recall@1, underscoring its applicability to real-world RCA challenges. Notably, CI-RCA and Shapley Flow emerge as close second-best performers, likely because both methods leverage causal rather than just correlational information, which appears to be advantageous in this context.

## 6.1 Ablation Study

Due to the page limit, we only present an overview of the major findings here. More details can be found in Appendix G.5- G.7. (i) **Impact of Label Proportions**: Incorporating even a modest amount (5-10%) of labeled data significantly enhances DeepITE's performance across various datasets, with recall improvements ranging from 4-20%. This approach provides substantial benefits in practical settings, unlike other baseline methods in the study that cannot utilize labeled data at all. (ii) **Replacement of Encoder and Decoder**: Modifying DeepITE's encoder to DAG-GNN's and its decoder to VACA's in two ablation designs shows that DeepITE–with its flexible generative and inference models–outperforms both modified versions and the original VACA [21] and DAG-GNN [15]. These observations, especially under conditions of increasing graph complexity, highlight

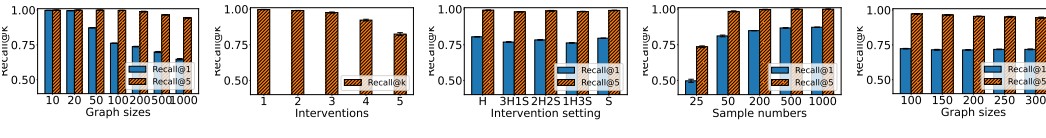

| (a) Graph sizes | (b) Interventions | (c) Mixed hard&soft | (d) Graph samples | (e) Mixed graphs |

Figure 2: The performance of DeepITE as a function of (a) graph sizes, (b) interventions, (c) the mixture proportion of soft and hard interventions, (d) sample size for each graph, (e) the number of mixed graphs.

DeepITE's robustness and the limitations of DAG-GNN's rigid design and VACA's requirement of minimal decoder layers in estimating ITE. (iii) **Scalability**: Figure 2(a-b) revealed the performance of DeepITE as a function of the graph size and the number of interventions respectively. Our findings indicate that while performance in terms of Recall@1 declines as the graph size increases, Recall@5 remains stable even for graphs with 1000 nodes—a size that is already considered quite large for causal analysis. (iv) **Hard&Soft Intervention**: We examined DeepITE under varying ratios of hard and soft interventions, where soft interventions were modeled by modifying the linear structural equations of the intervention targets to quadratic forms. Figure 2(c) shows DeepITE's adaptability to these mixtures, confirming its effectiveness in handling both types of interventions with only minimal performance loss. (v) **Samples**: Figure 2(d) illustrates the performance of DeepITE with variations in sample size. Our results show that DeepITE exhibits robustness with performance generally improving as the sample size increases. In contrast, traditional ITE methods [3, 6, 10] typically require thousands of samples for a single graph and intervention set to perform well. This resilience can be attributed to the collaborative learning framework of DeepITE and the relatively few parameters in the GNN-based encoder and decoder. (vi) **Amortization**: The performance of DeepITE as more graphs with different sizes are trained together, is detailed in Figure 2(e). Our findings indicate a minimal gradual degradation in the performance of DeepITE (mix) as we incorporate more graphs of varying sizes, attributed to the amortization error. Moreover, Table 1 shows that DeepITE (mix) even outperforms DeepITE (sep) training exclusively on 100-node graphs in terms of Recall@1. Based on this evidence, we maintain that the amortization process across graphs does not significantly hinder the performance.

## 6.2 Runtime Analysis

We conducted a runtime analysis using four distinct datasets, with variable counts $m$ ranging from 5 to 500, specifically $m = [5, 10, 11, 20, 50, 100, 500]$. For each value of $m$, we executed 10 trials on a set of 1000 samples and reported the average runtime. To ensure a fair comparison, we focused exclusively on the code pertinent to intervention identification. Timing commenced the moment the algorithm received the dataset and accompanying causal graph, if applicable, and ceased immediately upon delivery of the final results. This process ensured that our analysis exclusively measured the performance of the algorithm's core intervention-targeting functionality.

The runtime performance of the various methods, relative to graph size, is depicted in Appendix Figure 3. From the analysis, we note that DeepITE's runtime curve, represented in black, has the gentlest slope, implying that it boasts the lowest time complexity among all the methods. Notably, DeepITE secures the shortest runtime for graphs with more than 100 nodes. This heightened efficiency is attributable to DeepITE's inference process, which necessitates only a single pass through one branch of the inference network. In contrast, UT-IGSP exhibits the highest time complexity as it engages in an exponentially growing number of hypothesis tests to identify intervention targets. For instance, when handling graphs with $m = 500$ nodes, UT-IGSP requires nearly an hour to complete a single run.

## 7 Conclusion

In this paper, we presented DeepITE, a novel VGAE for ITE. By carefully design the VGAE based on GNNs, DeepITE allows collaborative learning and amortized inference across data with a range of intervention targets and causal graphs. The model adeptly supports both self-supervised and semi-supervised learning modalities, effectively harnessing labeled data to refine ITE. Our comprehensive results demonstrate that DeepITE can be seamlessly adapted to a multitude of domains, accommodating diverse causal graph configurations while exhibiting superior performance in terms of both Recall@k metrics and computational efficiency.

## Acknowledgements

We would like to thank Ant Group for their support for this work.

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

Table 3: Notations and their meanings.

| Notation | SIZE | Meaning |
|---|---|---|
| $\mathcal{G}$ | | causal graph |
| $\boldsymbol{A}$ | $m \times m$ | the asymmetric adjacency matrix of observational DAG |
| $\mathbf{x}$ | $m \times 1$ | endogenous variables |
| $\mathbf{u}$ | $m \times 1$ | exogenous variables |
| $\mathrm{An}(i)$ | | the ancestor nodes of node $i$ |
| $\mathbf{u}_{\mathrm{An}^*(i)}$ | | exogenous variables of the ancestor nodes of node $i$ |
| $\mathrm{do}(\mathbf{x}_i = x_i)$ | | do-operation |
| $\mathcal{I}$ | | set of the intervened nodes |
| $\boldsymbol{A}_{\mathcal{I}}$ | $m \times m$ | the adjacency matrix of interventional DAG |
| $\mathbf{1}$ | $m \times 1$ | a column vector of all ones |
| $\odot$ | | Hadamard product |
| $q(\cdot)$ | | variational q-distribution |
| $p(\cdot)$ | | variational p-distribution |
| $\mathcal{N}$ | | normal distribution |
| Bern | | Bernoulli distribution |
| Lognormal | | log-normal distribution |
| $\mathbf{u}$ | $m \times 1$ | latent variables for normal distribution |
| $\boldsymbol{\gamma}$ | $m \times 1$ | latent variables for Bernoulli distribution |
| $\epsilon$ | $1 \times 1$ | global observation noise |
| $\zeta$ | $1 \times 1$ | inverse variance of the noise |
| $\mu_i$ | $1 \times 1$ | mean of normal distribution of node $i$ |
| $\sigma_i^2$ | $1 \times 1$ | variance of normal distribution of node $i$ |
| $\pi$ | $1 \times 1$ | probaility of taking one in a Bernoulli distribution |
| $\mu_\zeta$ | $m \times 1$ | mean of log-normal distribution |
| $\sigma_\zeta^2$ | $m \times 1$ | variance of log-normal distribution |
| $\mu_i(\mathbf{x}, \boldsymbol{A})$ | $1 \times 1$ | mean of the estimated normal distribution of node $i$ |
| $\sigma_i^2(\mathbf{x}, \boldsymbol{A})$ | $1 \times 1$ | variance of the estimated Normal distribution of node $i$ |
| $\omega_i(\mathbf{x}, \boldsymbol{A})$ | $1 \times 1$ | estimated probaility of taking one in a Bernoulli distribution of node $i$ |
| $\mu_\zeta(\mathbf{x}, \boldsymbol{A})$ | $1 \times 1$ | mean of the estimated log-normal distribution |
| $\sigma_\zeta^2(\mathbf{x}, \boldsymbol{A})$ | $1 \times 1$ | variance of the estimated log-normal distribution |
| $p(\mathbf{x}, \mathbf{u}, \boldsymbol{\gamma}, \zeta | \boldsymbol{A})$ | | the proposed gernerative model |
| $q(\mathbf{u}, \boldsymbol{\gamma}, \zeta | \mathbf{x}, \boldsymbol{A})$ | | the proposed inference model |
| $\mathbb{E}_q$ | | expectation over the q-distribution |
| $\mathbb{H}_q$ | | entropy of the q-distribution |
| $\mathcal{L}$ | | the evidence lower bound (ELBO) |
| $D_{\mathrm{KL}}(q||p)$ | | KL divergence between distributions q and p |
| $\mathrm{Dec}(\mathbf{u}, \boldsymbol{\gamma}, \boldsymbol{A})$ | | the proposed decoder |

# A    Notations

See Table 3.

# B    More on Related Works

**Causal Explanations**: We also briefly review causal explanations as they are related to ITE. AI explainability endeavors to demystify model decisions, a pursuit encompassing feature attribution and contrastive explanations. Feature attribution methods initially focused on associations, revealing how input features correlate with predictions [24]. Moving beyond mere correlations, recent approaches integrate causality to enhance interpretability: CXPlain [32] employs supervised learning to discern the causal impact of features on model predictions, though computational demands escalate with

the need for repeated model evaluations. Generative Causal Explanations (GCE) [33] introduce disentangled latent factors to isolate causal effects, yet ensuring these factors accurately reflect the data distribution is complex. Asymmetric Shapley Value (ASV) [25], Shapley Flows [26], and PWSHAP [27] further this trend by considering the causal graphs, with the former adapting Shapley values to reflect causal structures and the latter two focusing on the causal relationships signified by the graph's edges and paths. On the other hand, Contrastive explanations, illustrated by Counterfactual Explanations (CE) [34] and Causal Algorithmic Recourse (CAR) [35, 36], offer a counterfactual narrative on how slight modifications or specific interventions could lead to different, often more favorable outcomes. Specifically, CE suggests minimal feature changes for an alternate outcome, teaching individuals how to achieve a different result. Their application must navigate constraints of plausibility and actionability to avoid recommending impractical changes. CAR extends CEs by proposing interventions grounded in causal relationships, aiming to recalibrate outcomes with consideration of the cost and effect of actions.

These causal explanation methods intersect with ITE. They can serve as a foundation for ITE by treating the most influential factors as potential intervention targets. Conversely, ITE can reciprocate by informing causal explanations since it pinpoints the very intervention targets that are the roots of observed outcomes.

**Root Cause Analysis**: Lastly, graph-based RCA methods warrant discussion, due to the intertwined nature of root causes and intervention targets. These methods usually operate in two distinct stages. Initially, the graph structure is established either via causal discovery algorithms such as the PC algorithm [28, 37, 38, 39, 40, 41, 42, 43], Granger causality [44] and DAG-GNN [31], or it is derived from domain-specific knowledge like topology graphs [45, 46, 29, 47, 48, 49, 30]. The second stage then leverages algorithms such as PageRank [46, 31], breadth-first or depth-first search [28, 37, 38, 29], and random walk [50, 40, 41, 42, 43, 30] for root cause localization within the graph. Despite their utility, the second stage tends to focus on association rather than causation, assigning a higher correlation to the connections between a node and its parents over those between the node and its children to consider the directionality in the causal graph. In contrast, as highlighted in the preceding sections, CI-RCA [4] emphasizes causation, utilizing linear regression-based hypothesis testing to pinpoint intervention targets. As an alternative, RCD [9] incorporates the $\Psi$-PC algorithm in a hierarchical fashion, intertwining the learning of intervention targets with graph structure discovery. These approaches not only improve root cause identification but also harmonize with ITE goals, fostering a causally informed analysis in RCA. However, as noted earlier, these methods lack collaborative learning and are fully unsupervised, necessitating complete inference for each RCA instance from scratch and failing to utilize labeling information.

## C    Proof of Proposition 1

According to the Neumann power series for the matrix inverse, we can obtain:

$$(\boldsymbol{I} - \boldsymbol{A}^T)^{-1} = \sum_{k=0}^{\infty} (\boldsymbol{A}^k)^T, \tag{15}$$

where $A^k$ denotes the $k$-th power of the matrix $A$, which involves multiplying the matrix $A$ by itself $k$ times. In the above expression, entry $(i, j)$ in the $k$-th power of $\boldsymbol{A}$ can be elaborated as:

$$(\boldsymbol{A}^k)_{ij} = \sum_{w_1, \ldots, w_{k-1}} \boldsymbol{A}_{i, w_1} \boldsymbol{A}_{w_1, w_2} \ldots \boldsymbol{A}_{w_{k-1}, j}$$

$$= \sum \mathrm{path}_k(i, j), \tag{16}$$

where the sum encapsulates all paths $(i, w_1, w_2, \ldots, w_{k-1}, j)$ from $i$ to $j$ with length $k$. In directed graphs, these paths must observe edge directionality. Consequently, for DAGs, the matrix $\boldsymbol{A}^k$ becomes a zero matrix when $k$ surpasses the graph diameter $\delta$, as no paths of length $k$ exist between any two nodes within such graphs. Substitute (15) into (5), and the GNN layer is recast as:

$$\mathbf{x} = f_2 \Big( \sum_{k=0}^{\infty} (\boldsymbol{A}^k)^T f_1(\mathbf{u}) \Big). \tag{17}$$

By multiplying $(\boldsymbol{A}^k)^T$ with $f_1(\mathbf{u})$ as in (17), node $i$ can receive information from its ancestors that have a path of length $k$ connecting to $i$. As $k$ goes from 0 to $\infty$, node $i$ accrues information from the

input features of its own and all its ancestors. On the other hand, paths only exist between node $i$ and one of its ancestors. Conversely, such paths are non-existent between node $i$ and non-ancestral nodes; hence, node $i$ exclusively assimilates inputs from its ancestors, thereby concluding the proof.

## D    Proof of Proposition 2

In consideration of Proposition 1 and given the decoder in Eq. (5), it is evident that $x_i$ is a function solely of $\mathbf{u}_{\mathrm{An}^*(i)}$ for all indices $i$. Consequently, the probability distribution $p(\mathbf{x}|\mathbf{u}, \boldsymbol{A})$ can be factorized into:

$$p(\mathbf{x}|\mathbf{u}, \boldsymbol{A}) = \prod_i p(x_i|\mathbf{u}_{\mathrm{An}^*(i)}), \tag{18}$$

which logically concludes the proof.

## E    Proof of Proposition 3

The essence of a causal intervention lies in severing all the incoming edges to the intervened nodes. Therefore, the decoder stipulated in Eq. (5) can faithfully represent causal interventions only if it encompasses all possible causal pathways; otherwise, severing certain pathways would exert no influence on the resulting intervention configuration. The decoder in Eq. (5) does indeed model all causally relevant paths, as corroborated by Proposition 1, thereby completing the proof.

## F    Derivation of the ELBO

Recall that the generative model (i.e., the $p$ distribution) can be factorized as:

$$p(\mathbf{x}, \mathbf{u}, \boldsymbol{\gamma}, \zeta|\boldsymbol{A}) = p(\mathbf{x}|\mathbf{u}, \boldsymbol{\gamma}, \boldsymbol{A}, \zeta)p(\zeta) \prod_{i=1}^m p(\mathbf{u}_i)p(\gamma_i), \tag{19}$$

where

$$p(\mathbf{u}_i) = \mathcal{N}(0, 1), \quad \forall i, \tag{20}$$

$$p(\gamma_i) = \mathrm{Bern}(\pi), \quad \forall i, \tag{21}$$

$$p(\zeta) \propto 1/\zeta, \tag{22}$$

$$p(\mathbf{x}|\mathbf{u}, \boldsymbol{\gamma}, \boldsymbol{A}, \zeta) = \mathcal{N}\big(\mathrm{Dec}(\mathbf{u}, \boldsymbol{\gamma}, \boldsymbol{A}), \zeta^{-1}\boldsymbol{I}\big). \tag{23}$$

, and the inference model (i.e., the $q$ distribution) can be factorized as:

$$q(\mathbf{u}, \boldsymbol{\gamma}, \zeta|\mathbf{x}, \boldsymbol{A}) = q(\zeta|\mathbf{x}, \boldsymbol{A}) \prod_{i=1}^m q(\mathbf{u}_i|\mathbf{x}, \boldsymbol{A})q(\gamma_i|\mathbf{x}, \boldsymbol{A}), \tag{24}$$

where

$$q(\mathbf{u}_i|\mathbf{x}, \boldsymbol{A}) = \mathcal{N}\big(\mu_i(\mathbf{x}, \boldsymbol{A}), \sigma_i^2(\mathbf{x}, \boldsymbol{A})\big), \tag{25}$$

$$q(\gamma_i|\mathbf{x}, \boldsymbol{A}) = \mathrm{Bern}\big(\omega_i(\mathbf{x}, \boldsymbol{A})\big), \tag{26}$$

$$q(\zeta|\mathbf{x}, \boldsymbol{A}) = \mathrm{Lognormal}\big(\mu_\zeta(\mathbf{x}, \boldsymbol{A}), \sigma_\zeta^2(\mathbf{x}, \boldsymbol{A})\big). \tag{27}$$

Note that the parameters of the above $q$ distributions are explicit functions of the given sample of the endogenous variables $\mathbf{x}$ and the adjacency matrix $\boldsymbol{A}$, which is parameterized by the GAT-based inference network. By substituting the $p$ (19) and $q$ distributions (24) into the ELBO (14), we can obtain:

$$\mathcal{L} = \mathbb{E}_q\Big[\log p(\mathbf{x}|\mathbf{u}, \boldsymbol{\gamma}, \boldsymbol{A}, \zeta) + \log p(\zeta) + \sum_{i=1}^m \big(\log p(\mathbf{u}_i) + \log p(\gamma_i)\big)\Big] + \mathbb{H}_q,$$

$$= \mathbb{E}_q\Big[\log p(\mathbf{x}|\mathbf{u}, \boldsymbol{\gamma}, \boldsymbol{A}, \zeta) + \log p(\zeta) - \log q(\zeta|\mathbf{x}, \boldsymbol{A}) +$$

$$\sum_{i=1}^m \big(\log p(\mathbf{u}_i) - \log q(\mathbf{u}_i|\mathbf{x}, \boldsymbol{A}) + \log p(\gamma_i) - \log q(\gamma_i|\mathbf{x}, \boldsymbol{A})\big)\Big],$$

$$= \mathbb{E}_q\big[\log p(\mathbf{x}|\mathbf{u}, \boldsymbol{\gamma}, \boldsymbol{A}, \zeta)\big] - D_{\mathrm{KL}}\big(q(\zeta|\mathbf{x}, \boldsymbol{A})\|p(\zeta)\big) -$$

$$\sum_{i=1}^{m} \Big( D_{\mathrm{KL}}\big(q(\mathbf{u}_i|\mathbf{x}, \boldsymbol{A})\|p(\mathbf{u}_i)\big) + D_{\mathrm{KL}}\big(q(\gamma_i|\mathbf{x}, \boldsymbol{A})\|p(\gamma_i)\big) \Big), \tag{28}$$

where $D_{\mathrm{KL}}$ denotes the KL (Kullback-Leibler) divergence between two distributions. We now delve into the expectation term in the above expression, which is given by:

$$\mathbb{E}_q\big[\log p(\mathbf{x}|\mathbf{u}, \boldsymbol{\gamma}, \boldsymbol{A}, \zeta)\big]$$
$$= \frac{m}{2}\langle\log\zeta\rangle - \frac{1}{2}\langle\zeta\rangle\langle(\mathbf{x} - \mathrm{Dec}(\mathbf{u}, \boldsymbol{\gamma}, \boldsymbol{A}))^T(\mathbf{x} - \mathrm{Dec}(\mathbf{u}, \boldsymbol{\gamma}, \boldsymbol{A}))\rangle,$$
$$= \frac{m}{2}\mu_\zeta(\mathbf{x}, \boldsymbol{A}) - \frac{1}{2}\exp\Big(\mu_\zeta(\mathbf{x}, \boldsymbol{A}) + \frac{\sigma_\zeta^2(\mathbf{x}, \boldsymbol{A})}{2}\Big)\cdot$$
$$\langle(\mathbf{x} - \mathrm{Dec}(\mathbf{u}, \boldsymbol{\gamma}, \boldsymbol{A}))^T(\mathbf{x} - \mathrm{Dec}(\mathbf{u}, \boldsymbol{\gamma}, \boldsymbol{A}))\rangle. \tag{29}$$

The remaining three KL divergence terms can be written as:

$$D_{\mathrm{KL}}\big(q(\zeta|\mathbf{x}, \boldsymbol{A})\|p(\zeta)\big) = -\frac{1}{2}\log\sigma_\zeta^2(\mathbf{x}, \boldsymbol{A}), \tag{30}$$

$$D_{\mathrm{KL}}\big(q(\mathbf{u}_i|\mathbf{x}, \boldsymbol{A})\|p(\mathbf{u}_i)\big) = \frac{1}{2}\big(\mu_i^2(\mathbf{x}, \boldsymbol{A}) + \sigma_i^2(\mathbf{x}, \boldsymbol{A}) - \log\sigma_i^2(\mathbf{x}, \boldsymbol{A})\big), \tag{31}$$

$$D_{\mathrm{KL}}\big(q(\gamma_i|\mathbf{x}, \boldsymbol{A})\|p(\gamma_i)\big) = \omega_i(\mathbf{x}, \boldsymbol{A})\,\mathrm{logit}(\omega_i(\mathbf{x}, \boldsymbol{A})) - \omega_i(\mathbf{x}, \boldsymbol{A})\,\mathrm{logit}(\pi)$$
$$+ \log(1 - \omega_i(\mathbf{x}, \boldsymbol{A})) - \log(1 - \pi). \tag{32}$$

---

**Algorithm 1** DeepITE Semi-supervised Training Algorithm

---

**Require:** Causal graph $\mathcal{G}$, adjacency matrix $\boldsymbol{A}$, endogenous variables $\mathbf{x}^{\{1:m\}}$, labels of intervention targets $\boldsymbol{\gamma}_{\mathrm{true}}^{\{1:m\}}$ if available.
**Ensure:** Parameters of the inference network $\phi$, parameters of the generative network $\theta$;
 1: Initialize $\phi, \theta$ randomly;
 2: **repeat**
 3:   Pass $\mathbf{x}^{\{1:m\}}$ and $\boldsymbol{A}$ through the inference network to get the parameters for $q(\mathbf{u}|\mathbf{x}^{\{1:m\}}, \boldsymbol{A})$, $q(\boldsymbol{\gamma}|\mathbf{x}^{\{1:m\}}, \boldsymbol{A}), q(\zeta|\mathbf{x}^{\{1:m\}}, \boldsymbol{A})$;
 4:   Draw samples from the Normal distribution $q(\mathbf{u}|\mathbf{x}^{\{1:m\}}, \boldsymbol{A})$;
 5:   Draw samples from the Bernoulli distribution $q(\boldsymbol{\gamma}|\mathbf{x}^{\{1:m\}}, \boldsymbol{A})$ using the gumbel-softmax reparameterization trick;
 6:   Draw samples from the Log Normal distribution $q(\zeta|\mathbf{x}^{\{1:m\}}, \boldsymbol{A})$;
 7:   $\widehat{\mathbf{x}}^{\{1:m\}} \leftarrow \mathrm{Dec}(\boldsymbol{u}^{\{1:m\}}, \boldsymbol{\gamma}^{\{1:m\}}, \zeta, \boldsymbol{A})$
 8:   **if** $\boldsymbol{\gamma}_{\mathrm{true}}^{\{1:m\}}$ is available **then**
 9:     Compute the negative ELBO (28) and the maximum log likelihood of $\boldsymbol{\gamma}_{\mathrm{true}}^{\{1:m\}}$;
10:   **else**
11:     Compute the negative ELBO (28);
12:   **end if**
13:   Update $\phi, \theta$ via gradient descent;
14: **until** convergence
15: **return** $\phi, \theta$

---

# G   Experiment Details

## G.1   Experiment Setup

Unless otherwise specified, in all of our experiments for DeepITE, we set the hidden dimension in GAT and MLP to 64. For optimization, we used NAdam [52] with a learning rate $1 \times 10^{-4}$. We conducted training for 1000 epochs and select the checkpoints with the lowest training loss. The temperature $\mathbf{t}$ for gumbel-softmax [23] is calculated by $\mathbf{t} = 101 - 0.2\mathbf{e}$ when epoch $\mathbf{e} <= 500$ and $\mathbf{t} = 0.5/(e - 500)$ for epoch $\mathbf{e} > 500$. All the training runs on 4 NVIDIA TESLA P100 GPUs with 50GB of VRAM. All the inference runs on a MacBook Pro 16 inch with a 6-core Intel i7 CPU and 16 GB of RAM.

The key assumptions and characteristics of the comparative methods, as well as the time complexity in the inference process, are summarized in the table 4 and table 5 To facilitate a fair comparison, all methods (including ITE, RCA, and XAI) are provided with the same ground truth causal graph,

Table 4: Key assumptions and characteristics of comparing methods. Here, $m$ and $n$ denote the number of nodes and edges in the DAG, $p_\Delta$ is the number of intervention targets given by the precision different estimation algorithm, and finally, $T$, $L$, and $D$ represent the number of trees, the depth of the trees, and the number of leaf nodes in the gradient-boosted trees.

| METHOD | CAUSAL GRAPH | REFERENCE SET | INTERVENTION | CONFOUNDER | AMORTIZATION | GRAPH SIZE | TIME COMPLEXITY |
|---|---|---|---|---|---|---|---|
| DeepITE | Given | No | Soft&Hard | No | Yes | < 1000 | $\mathcal{O}(m+n)$ |
| UT-IGSP | Unknown | Require | Soft | No | No | < 100 | $\mathcal{O}(2^{m-1})$ |
| CITE | Given | Require | Soft | No | No | < 100 | $\mathcal{O}(2^{p_\Delta})$ |
| PreDITEr | Given | Require | Soft | No | Yes | < 100 | $\mathcal{O}(2^{p_\Delta})$ |
| LIT | Unknown | Require | Soft | No | Yes | < 100 | $\mathcal{O}(m^2)$ |
| CauseInfer | Given | No | Hard | No | Yes | < 1000 | $\mathcal{O}(m^2)$ |
| MicroHECL | Given | No | Hard | No | Yes | < 1000 | $\mathcal{O}(m^2)$ |
| MicroRCA | Given | No | Hard | No | Yes | < 1000 | $\mathcal{O}(m^2) + O(m+n)$ |
| CausalRCA | Given | No | Hard | No | Yes | < 1000 | $\mathcal{O}(m^2)$ |
| CI-RCA | Given | No | Hard | No | Yes | < 1000 | $\mathcal{O}(m^2)$ |
| RCD | Given | No | Hard | No | Yes | < 1000 | $\mathcal{O}(m^3)$ |
| TreeExplainer | Unknown | No | Soft | No | No | < 100 | $\mathcal{O}(TLD^2)$ |
| ASV | Given | No | Soft | No | No | < 100 | $\mathcal{O}(m^3)$ |
| ShapleyFlow | Given | No | Soft | No | No | < 100 | $\mathcal{O}(m^3)$ |
| PWSHAP | Given | No | Soft | Yes | No | < 100 | $\mathcal{O}(m^3)$ |

Table 5: Time Complexity during Inference. Here, $m$ and $n$ denote the number of nodes and edges in the DAG, $p_\Delta$ is the number of intervention targets given by the precision different estimation algorithm, and finally, $T$, $L$, and $D$ represent the number of trees, the depth of the trees, and the number of leaf nodes in the gradient-boosted trees.

| METHOD | TIME COMPLEXITY |
|---|---|
| UT-IGSP [10] | $\mathcal{O}(2^{m-1})$ |
| CITE [3] | $\mathcal{O}(2^{p_\Delta})$ |
| PreDITEr [6] | $\mathcal{O}(2^{p_\Delta})$ |
| CauseInfer [28] | $\mathcal{O}(m^2)$ |
| MicroHECL [29] | $\mathcal{O}(m^2)$ |
| MicroRCA [30] | $\mathcal{O}(m^2) + O(m+n)$ |
| CausalRCA [31] | $\mathcal{O}(m^2)$ |
| CI-RCA [4] | $\mathcal{O}(m^2)$ |
| RCD [9] | $\mathcal{O}(m^3)$ |
| TreeExplainer [51] | $\mathcal{O}(TLD^2)$ |
| ASV [25] | $\mathcal{O}(m^3)$ |
| ShapleyFlow [26] | $\mathcal{O}(m^3)$ |
| PWSHAP [27] | $\mathcal{O}(m^3)$ |
| DeepITE | $\mathcal{O}(m+n)$ |

eschewing the need for graph construction from data for some RCA methods. Within the realm of XAI, constructing a forward predictive model is a prerequisite for backward attribution analysis—the process used to identify the intervention targets. To facilitate this, we construct a predictive model using gradient-boosted trees. This forward model is trained to predict whether the graph is intervened based on the full set of node features $\mathbf{x}$, in a supervised manner. During inference, we extract the intervention targets by identifying the top $k$ nodes that yield the highest attribution scores.

## G.2 Synthetic Data Generation

The generation process begins by creating a random adjacency matrix $\boldsymbol{A}$, which is structured to be upper-triangular to ensure the resulting DAG is indeed acyclic. The matrix's non-zero entries are uniformly distributed across the range $[-2, -0.5] \cup [0.5, 2]$, representing the possible strengths of causal relationships between nodes. For each node within the DAG, time series data are synthesized according to a model that captures causal dependencies, as inspired by the linear decoder equation: $\mathbf{x}(t) = (\boldsymbol{I} - \boldsymbol{A}^T)^{-1}(\mathbf{u}(t) + \beta\mathbf{x}(t-1))$, where $t$ indicates the discrete time steps, and $\beta$ denotes the autoregressive coefficient, influencing the temporal consistency of the data.

Interventions are then introduced in the time series at time $t$, with the set of intervention nodes $\mathcal{I}$ being randomly selected from all non-root nodes, and the size of $\mathcal{I}$ adhering to a Poisson distribution. For nodes within $\mathcal{I}$, we augment the corresponding exogenous noise, adhering to the three-sigma rule for significant deviation. Each generated time series spans 1000 time steps, with interventions

Table 6: Recall@k of different algorithms for detecting the intervened nodes from the Synthetic dataset following the classical ITE setting.

| DATASET | Linear-10 | | Linear-20 | | Nonlinear-10 | | Nonlinear-20 | |
|---|---|---|---|---|---|---|---|---|
| METRICS | Recall@1 | Recall@3 | Recall@1 | Recall@3 | Recall@1 | Recall@3 | Recall@1 | Recall@3 |
| UT-IGSP | $0.921 \pm 0.009$ | $0.972 \pm 0.005$ | $0.917 \pm 0.007$ | $0.956 \pm 0.005$ | $0.933 \pm 0.008$ | $0.989 \pm 0.004$ | $0.927 \pm 0.005$ | $0.972 \pm 0.004$ |
| CITE | $0.939 \pm 0.007$ | $0.989 \pm 0.003$ | $0.933 \pm 0.009$ | $0.961 \pm 0.004$ | $0.789 \pm 0.011$ | $0.956 \pm 0.006$ | $0.733 \pm 0.010$ | $0.883 \pm 0.005$ |
| PreDITEr | $0.944 \pm 0.006$ | $0.989 \pm 0.003$ | $0.933 \pm 0.008$ | $0.967 \pm 0.006$ | $0.694 \pm 0.005$ | $0.861 \pm 0.004$ | $0.561 \pm 0.004$ | $0.822 \pm 0.003$ |
| TreeExplainer | $0.764 \pm 0.015$ | $0.897 \pm 0.020$ | $0.551 \pm 0.010$ | $0.809 \pm 0.018$ | $0.765 \pm 0.015$ | $0.909 \pm 0.021$ | $0.539 \pm 0.010$ | $0.773 \pm 0.017$ |
| ASV | $0.751 \pm 0.014$ | $0.821 \pm 0.019$ | $0.629 \pm 0.011$ | $0.782 \pm 0.017$ | $0.679 \pm 0.014$ | $0.737 \pm 0.018$ | $0.510 \pm 0.010$ | $0.554 \pm 0.016$ |
| ShapleyFlow | $0.858 \pm 0.017$ | $0.928 \pm 0.022$ | $0.817 \pm 0.012$ | $0.858 \pm 0.019$ | $0.841 \pm 0.016$ | $0.919 \pm 0.023$ | $0.811 \pm 0.011$ | $0.870 \pm 0.020$ |
| PWSHAP | $0.784 \pm 0.016$ | $0.892 \pm 0.021$ | $0.650 \pm 0.011$ | $0.815 \pm 0.018$ | $0.776 \pm 0.014$ | $0.895 \pm 0.015$ | $0.647 \pm 0.010$ | $0.781 \pm 0.014$ |
| CauseInfer | $0.869 \pm 0.001$ | $0.935 \pm 0.001$ | $0.430 \pm 0.003$ | $0.615 \pm 0.003$ | $0.720 \pm 0.002$ | $0.819 \pm 0.001$ | $0.561 \pm 0.004$ | $0.692 \pm 0.003$ |
| MicroHECL | $0.407 \pm 0.011$ | $0.624 \pm 0.012$ | $0.387 \pm 0.014$ | $0.485 \pm 0.019$ | $0.420 \pm 0.012$ | $0.660 \pm 0.015$ | $0.411 \pm 0.017$ | $0.536 \pm 0.022$ |
| MicroRCA | $0.759 \pm 0.002$ | $0.890 \pm 0.001$ | $0.730 \pm 0.003$ | $0.846 \pm 0.002$ | $0.398 \pm 0.002$ | $0.527 \pm 0.002$ | $0.175 \pm 0.002$ | $0.233 \pm 0.002$ |
| CausalRCA | $0.820 \pm 0.003$ | $0.885 \pm 0.002$ | $0.705 \pm 0.002$ | $0.841 \pm 0.001$ | $0.729 \pm 0.003$ | $0.805 \pm 0.003$ | $0.545 \pm 0.002$ | $0.608 \pm 0.001$ |
| CI-RCA | $0.865 \pm 0.002$ | $0.941 \pm 0.001$ | $0.734 \pm 0.003$ | $0.923 \pm 0.003$ | $0.821 \pm 0.003$ | $0.896 \pm 0.002$ | $0.616 \pm 0.002$ | $0.683 \pm 0.002$ |
| RCD | $0.803 \pm 0.004$ | $0.871 \pm 0.005$ | $0.695 \pm 0.005$ | $0.837 \pm 0.007$ | $0.713 \pm 0.005$ | $0.801 \pm 0.006$ | $0.540 \pm 0.006$ | $0.597 \pm 0.006$ |
| DeepITE | $0.999 \pm 0.000$ | $\mathbf{1.000} \pm 0.000$ | $\mathbf{1.000} \pm 0.000$ | $\mathbf{1.000} \pm 0.000$ | $0.998 \pm 0.001$ | $\mathbf{1.000} \pm 0.000$ | $\mathbf{0.999} \pm 0.000$ | $\mathbf{0.999} \pm 0.000$ |

occurring randomly between time steps 100 and 900. Data preceding the intervention time $t$ are labeled as observational, while data from $t$ onwards are categorized as interventional. We explore causal graphs of varying complexity, with 50, 100, and 500 nodes, and associated edge counts of 100, 500, and 5000, respectively. To ensure a comprehensive assessment, 10 unique graphs are generated for each size, and for each graph, we create 100 distinct instances with varying intervention targets. The dataset is partitioned with an 85:5:10 ratio for training, validation, and testing.

## G.3 Synthetic Data under the Classical ITE Setting

For an equitable evaluation with existing ITE approaches, we further devise experiments on synthetically generated datasets tailored to classical ITE configurations, using dataset generation scripts from VACA [21]. Our setup included two linear and two non-linear SCMs. Initially, we generate 10,000 observational samples from each SCM. Subsequently, we perform ten distinct interventions on randomly selected variables, mimicking the procedures followed by ITE methods. Each intervention resulted in the creation of 1,000 samples. Throughout these interventions, we maintain the SCMs' modularity and provid the causal graph with the data. We divide the dataset into training, validation, and testing batches with an 85:5:10 ratio, respectively.

Table 6 reveals that DeepITE consistently surpasses competing methods in performance. Notably, traditional ITE approaches like UT-IGSP, CITE, and PreDITEr benefit significantly from an adequate pool of observational samples on small-scale graphs, reflecting a marked improvement in their effectiveness. Similarly, XAI techniques—ShapleyFlow in particular—demonstrate commendable performance, proving to be well-adapted for smaller graphs where their capabilities can be fully leveraged. Conversely, RCA methods exhibit weaker results, a trend that may be linked to their design inclination towards larger-scale graphs. These larger configurations are often seen in complex environments such as microservice diagnosis, suggesting that RCA methods' strengths are not fully utilized in smaller or less complex scenarios highlighted in this comparison.

Furthermore, it is evident that models based on the premise of linear SCMs, such as CITE, PreDITEr, and CI-RCA, tend to underperform in settings that demand an understanding of non-linear dynamics. In contrast, DeepITE, which utilizes learnable SCMs, demonstrates impressive versatility by effectively addressing both linear and non-linear scenarios.

## G.4 Real Data Description

**Protein Signaling Dataset**: The well-known protein signaling dataset, which originates from Sachs *et al.* [53], investigates the complex interactions within T-4 cell signaling networks. The dataset comprises 11 nodes and 16 edges, with a collection of 1755 observational and 4091 interventional samples derived from five different experimental environments where various drugs were used to modulate signaling proteins. We harness an accepted ground truth network structure in [54], and the preprocessing steps outlined in [10], to benchmark DeepITE's performance with other models. The dataset is partitioned with an 85:5:10 ratio for training, validation, and testing.

Table 7: Impact of Labeled Data on DeepITE (mix) for the Synthetic Dataset. The proportion of labeled data is shown in the brackets.

| DATASET | HeterITE-50 | | HeterITE-100 | | HeterITE-500 | |
|---|---|---|---|---|---|---|
| METRICS | Recall@1 | Recall@5 | Recall@1 | Recall@5 | Recall@1 | Recall@5 |
| DeepITE (0%) | $0.718 \pm 0.004$ | $0.945 \pm 0.001$ | $0.690 \pm 0.004$ | $0.923 \pm 0.002$ | $0.627 \pm 0.004$ | $0.875 \pm 0.002$ |
| DeepITE (5%) | $0.820 \pm 0.002$ | $0.986 \pm 0.002$ | $0.731 \pm 0.003$ | $0.960 \pm 0.001$ | $0.667 \pm 0.003$ | $0.922 \pm 0.002$ |
| DeepITE (10%) | $0.821 \pm 0.002$ | $0.991 \pm 0.001$ | $0.728 \pm 0.003$ | $0.944 \pm 0.003$ | $0.674 \pm 0.003$ | $0.925 \pm 0.002$ |
| DeepITE (25%) | $0.829 \pm 0.002$ | $0.995 \pm 0.001$ | $0.737 \pm 0.002$ | $0.982 \pm 0.001$ | $0.672 \pm 0.004$ | $0.954 \pm 0.001$ |
| DeepITE (50%) | $0.856 \pm 0.002$ | $0.998 \pm 0.000$ | $0.750 \pm 0.003$ | $0.996 \pm 0.001$ | $0.678 \pm 0.003$ | $0.952 \pm 0.002$ |
| DeepITE (75%) | $\mathbf{0.873} \pm 0.001$ | $\mathbf{1.000} \pm 0.000$ | $\mathbf{0.762} \pm 0.003$ | $\mathbf{0.998} \pm 0.000$ | $\mathbf{0.699} \pm 0.004$ | $\mathbf{0.966} \pm 0.002$ |
| DeepITE (100%) | $0.869 \pm 0.002$ | $0.999 \pm 0.001$ | $0.773 \pm 0.002$ | $\mathbf{0.998} \pm 0.001$ | $0.697 \pm 0.002$ | $\mathbf{0.966} \pm 0.001$ |

Table 8: Impact of Labeled Data on DeepITE for the two Real Datasets. The proportion of labeled data is shown in the brackets.

| DATASET | Protein Signaling | ICASSP-SPGC 2022 | | | |
|---|---|---|---|---|---|
| METRICS | Recall@1 | Recall@1 | Recall@5 | Root.Acc | Score |
| DeepITE (0%) | $0.652 \pm 0.004$ | $0.881 \pm 0.002$ | $0.984 \pm 0.001$ | $0.9794 \pm 0.0054$ | $0.9085 \pm 0.0101$ |
| DeepITE (5%) | $0.842 \pm 0.002$ | $0.906 \pm 0.001$ | $0.994 \pm 0.001$ | $\mathbf{0.9964} \pm 0.0023$ | $\mathbf{0.9524} \pm 0.0040$ |
| DeepITE (10%) | $0.850 \pm 0.003$ | $0.920 \pm 0.001$ | $0.994 \pm 0.001$ | $\mathbf{0.9964} \pm 0.0000$ | $\mathbf{0.9524} \pm 0.0000$ |
| DeepITE (25%) | $0.849 \pm 0.002$ | $0.922 \pm 0.001$ | $0.994 \pm 0.000$ | $\mathbf{0.9964} \pm 0.0000$ | $\mathbf{0.9524} \pm 0.0000$ |
| DeepITE (50%) | $0.868 \pm 0.002$ | $\mathbf{0.925} \pm 0.000$ | $\mathbf{0.995} \pm 0.000$ | $\mathbf{0.9964} \pm 0.0000$ | $\mathbf{0.9524} \pm 0.0000$ |
| DeepITE (75%) | $0.863 \pm 0.002$ | $0.924 \pm 0.001$ | $\mathbf{0.995} \pm 0.000$ | $\mathbf{0.9964} \pm 0.0000$ | $\mathbf{0.9524} \pm 0.0000$ |
| DeepITE (100%) | $\mathbf{0.872} \pm 0.001$ | $\mathbf{0.925} \pm 0.000$ | $\mathbf{0.995} \pm 0.000$ | $\mathbf{0.9964} \pm 0.0000$ | $\mathbf{0.9524} \pm 0.0000$ |

**ICASSP-SPGC 2022**[5]: The ICASSP-SPGC 2022 dataset [55], derived from active 5G networks, is a real-world telecommunications dataset for RCA comprising 2984 samples and 23 variables that represent various Key Performance Indicators (KPIs). Human experts have verified the accompanying causal graph, yet only around 45% of the data are explicitly labeled with root cause faults. Unlike the previously mentioned dataset, where interventions are directly known from labels, the root causes here are represented by unobserved variables outside the causal graph. However, experts provide a mapping of each root cause to observable causal variables. During training with labeled data, we treat these associated variables as if they had been intervened upon. Additionally, 600 extra samples are provided for testing purposes.

## G.5 Semi-Supervised Learning

In this section, we delve into the influence of labeled data on the efficacy of DeepITE, with our findings summarized in Tables 7-8. Across all datasets under consideration, it is clear that the incorporation of labeled data yields a substantial enhancement in DeepITE's performance, with improvements in Recall@1 ranging between 4% to 20%, depending on the dataset. Notably, the most pronounced gains are observed when the initial 5% to 10% of labeled data are integrated, with the rate of improvement tapering off beyond this point. This suggests that even a modest quantity of labeled data can lead to significant performance boosts, a fact that bears particular relevance in practical scenarios where acquiring a limited amount of labeled data is typically feasible and can offer considerable benefits to DeepITE. Conversely, the other baseline methods in our study do not possess the capability to leverage such labeling information to their advantage.

## G.6 Ablation Study

The ablation study of DeepITE, in comparison to VACA and DAG-GNN, is presented in Table 9. In Section 5.2, we have delved into the relationships between DeepITE, DAG-GNN, and VACA. We highlighted the limitations of DAG-GNN's inference model and VACA's generative model and illustrated how DeepITE overcomes these shortcomings. DAG-GNN utilizes an inference model represented as $\mathbf{u} = f_4((\boldsymbol{I} - \boldsymbol{A}^T) f_3(\mathbf{x}))$, which has constraints as it can only gather messages from a node's parents. On the other hand, DeepITE's inference model is crafted to aggregate messages from all nodes within the Markov blanket of a given node, ensuring a more flexible inference model tailored

---

[5]https://www.aiops.sribd.cn/home/statement

Table 9: Ablation study

| | | MMD(Obs) | MMD(Int) | SSE |
|---|---|---|---|---|
| | VACA | $3.90 \pm 0.17$ | $59.3 \pm 5.3$ | $3742.46 \pm 495.35$ |
| | DAG-GNN | $4.47 \pm 0.26$ | $67.80 \pm 7.14$ | $4525.58 \pm 539.04$ |
| Graph-50 | DeepITE (VACA Decoder) | $1.93 \pm 0.38$ | $10.78 \pm 2.99$ | $2534.70 \pm 105.22$ |
| | DeepITE (DAG-GNN Encoder) | $2.58 \pm 0.40$ | $32.80 \pm 7.89$ | $3177.97 \pm 189.61$ |
| | DeepITE | $\mathbf{0.12} \pm 0.075$ | $\mathbf{0.76} \pm 0.051$ | $\mathbf{415.66} \pm 26.92$ |
| | VACA | $4.58 \pm 0.26$ | $70.89 \pm 4.14$ | $4333.90 \pm 584.71$ |
| | DAG-GNN | $5.41 \pm 0.30$ | $89.71 \pm 6.33$ | $5410.52 \pm 656.23$ |
| Graph-100 | DeepITE (VACA Decoder) | $2.14 \pm 0.15$ | $14.7 \pm 1.53$ | $2967.47 \pm 151.69$ |
| | DeepITE (DAG-GNN Encoder) | $3.13 \pm 0.29$ | $73.92 \pm 5.96$ | $3651.14 \pm 245.47$ |
| | DeepITE | $\mathbf{0.19} \pm 0.073$ | $\mathbf{0.96} \pm 0.066$ | $\mathbf{490.06} \pm 37.15$ |
| | VACA | $5.36 \pm 0.54$ | $116.30 \pm 2.59$ | $6846.90 \pm 795.12$ |
| | DAG-GNN | $7.39 \pm 0.46$ | $154.96 \pm 3.63$ | $7511.26 \pm 916.93$ |
| Graph-500 | DeepITE (VACA Decoder) | $3.48 \pm 0.16$ | $31.89 \pm 5.08$ | $4391.59 \pm 340.15$ |
| | DeepITE (DAG-GNN Encoder) | $4.61 \pm 0.22$ | $48.60 \pm 8.20$ | $5316.33 \pm 454.54$ |
| | DeepITE | $\mathbf{1.08} \pm 0.045$ | $\mathbf{5.16} \pm 0.65$ | $\mathbf{731.17} \pm 75.32$ |

for ITE tasks. VACA employs a generative model with a minimum requirement of $\delta - 1$ MPNN layers, where $\delta$ represents the graph diameter. This limitation restricts the propagation distance of information within the graph, hindering its performance in estimating distributions over large graphs. DeepITE distinguishes itself by employing the generative model, specified in Eq. (7), thereby overcomes the limitations imposed by VACA's dependence on graph diameter.

To demonstrate the superiority of DeepITE's generative and inference models, we devised two ablation designs by seperately modifying DeepITE's encoder layers to DAG-GNN's encoder and DeepITE's decoder layers to VACA's decoder. These settings were compared alongside VACA and DAG-GNN. Based on the synthetic data in Appendix G.2, we combined the observational and interventional data and fed them into the model along with the adjacency matrix $\boldsymbol{A}$ for observational data, which is exactly how ITE works. Since VACA and DAG-GNN do not directly output ITE results, we utilized Maximum Mean Discrepancy (MMD) and the standard deviation of the squared error (SSE) between the true and estimated values for our evalutaion, providing another dimension to gauge their effectiveness in estimating ITE. MMD was calculated separately for observational and interventional data, even though this information was unknown to the models.

The results, as shown in Table 9, revealed DeepITE outperforming the other models, validating the excellence of DeepITE's generative and inference models. Due to its relatively inflexible model design, DAG-GNN struggles to effectively reconstruct the biases associated with intervention points. The constraint of minimal number decoder layers limits VACA's capability in capturing long-range dependencies and interactions within the graph structure, leading to a significant decrease in its performance as the number of graph nodes increase. Although the ablation methods showed suboptimal performance, the flexibility introduced by the ITE indicator $\gamma$ enabled them to outperform the origin VACA and DAG-GNN. This underscores the adaptability of our approach for ITE tasks despite its limitations in certain scenarios.

## G.7 Case Study

To address the challenges of root cause analysis in complex systems with interconnected variables, we conducted a case study on the real-world dataset ICASSP-SPGC 2022, evaluating the performance of DeepITE in comparison to other methods. The groundtruth causal graph is provided in [55]. Recall that we focus on the RCA problem and we aim to identify and present the root causes of the system to users. Note that while we have labels for the root causes in our testing data, we only possess observations for the observable variables represented in the graph. Consequently, all methodologies employed can only localize observable variables as intervention targets (ITs), rather than directly identifying the root causes. For instance, RootCause 2 (a weak signal in marginal areas) can influence feature19, feature X, and feature Y. However, RootCause 3 also affects feature X. As a result, when

Table 10: Case study on three different samples (1758, 1760, and 1093) from the real-world dataset ICASSP-SPGC 2022. ITE Top-k represents the k identified features. The root causes are then inferred from these k features. The ground truths for these samples are rootcause2, rootcause3, and rootcause2&rootcause3.

| | SAMPLE 1758 | | SAMPLE 1760 | | SAMPLE 1093 | |
| | ITE Top-1 | ROOT CAUSE | ITE Top-1 | ROOT CAUSE | ITE Top-2 | ROOT CAUSE |
|---|---|---|---|---|---|---|
| CauseInfer | FeatureY | Unspecified | FeatureY | Unspecified | Feature2&FeatureX | Unspecified |
| MicroHECL | FeatureX | Unspecified | FeatureY | Unspecified | FeatureX&Feature60 | RootCause3 |
| MicroRCA | FeatureY | Unspecified | FeatureY | Unspecified | FeatureX&Feature60 | RootCause3 |
| CausalRCA | FeatureY | Unspecified | FeatureX | Unspecified | FeatureY&FeatureX | Unspecified |
| CI-RCA | FeatureY | Unspecified | FeatureX | Unspecified | FeatureX&Feature60 | RootCause3 |
| RCD | FeatureY | Unspecified | FeatureX | Unspecified | FeatureY&FeatureX | Unspecified |
| TreeExplainer | FeatureX | Unspecified | Feature1 | Unspecified | FeatureX&Feature1 | Unspecified |
| ASV | FeatureY | Unspecified | FeatureX | Unspecified | FeatureY&FeatureX | Unspecified |
| ShapleyFlow | FeatureY | Unspecified | Feature17 | Unspecified | FeatureY&FeatureX | Unspecified |
| PWSHAP | FeatureY | Unspecified | FeatureX | Unspecified | FeatureX&Feature2 | Unspecified |
| DeepITE | Feature19 | RootCause2 | Feature60 | RootCause3 | Feature60&Feature19 | RootCause2&RootCause3 |

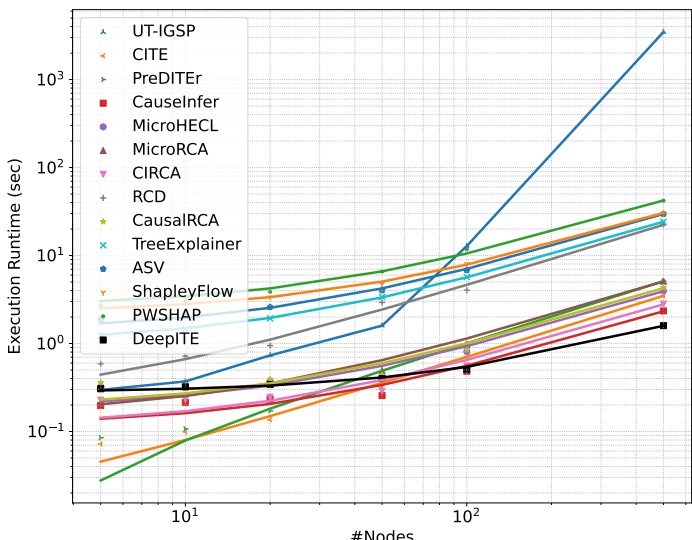

Figure 3: Runtime Analysis.

a given method identifies either feature X as an IT, it becomes challenging to ascertain whether RootCause 2 is indeed the true root cause.

Interestingly, DeepITE demonstrates superior performance on this dataset by effectively identifying features exclusive to a specific root cause. We evaluated the performance of all methods on samples 1758, 1760, and 1093, and summarized the results in Table 10. For sample 1758, where the true root cause is RootCause 2, DeepITE identifies feature19 as the IT, thus it is evident that RootCause2 should be the root cause. In contrast, other methods identify either feature X or feature Y, making it difficult to pinpoint the true root cause. Similarly, for sample 1760, DeepITE identifies feature60 as the IT, which is also exclusive to the true root cause, RootCause 3. On the other hand, in the case of sample 1093, DeepITE selects both feature60 and feature19, enabling us to conclude that both RootCause 2 and RootCause 3 are relevant root causes, a finding that aligns with the ground truth.

## H    Limitations

One limitation of the DeepITE framework is its applicability restricted to cases with fully observed causal graphs, presuming the absence of confounders. Real-world scenarios may involve confounding, where relationships between observed variables are influenced by latent variables. Addressing this challenge—how to effectively handle confounding in the presence of unobserved factors—represents a compelling avenue for future research. Additionally, DeepITE presupposes the availability of a pre-specified graph structure. While causal discovery techniques can be applied to ascertain the graph

configuration when it is not known, the joint pursuit of determining both the graph structure and the intervention targets simultaneously offers a tantalizing challenge for future exploration. Finally, proving the consistency and identifiability of DeepITE, and more broadly in the application of VGAEs for ITE, remains an interesting avenue for future work. Notably, such theoretical guarantees have been established for VGAEs in the context of causal inference (both observational and interventional) in [56].

