# OpenReview forum: "DeepITE: Designing Variational Graph Autoencoders for Intervention Target Estimation"
_NeurIPS.cc/2024/Conference — NeurIPS 2024 poster_

### Official Review · Reviewer_HUbk · 2024-06-25

**Soundness:** 3
**Presentation:** 3
**Contribution:** 3
**Rating:** 5
**Confidence:** 3

**Summary:**

The paper introduces a deep learning framework for identifying intervention targets in causal systems by amortizing the problem, although assuming that the causal graph is known. The framework, called DeepITE, employs a variational graph autoencoder (VGAE) that learns from both labeled and unlabeled data across different intervention targets and causal graphs. The model can quickly identify intervention targets in new samples without the need for retraining, addressing the inefficiencies of current methods which require frequent re-estimations with minor data changes. DeepITE is validated through comprehensive testing against 13 baseline methods, showing superior performance.

**Strengths:**

Originality
The proposed variational graph autoencoder (VGAE) framework tailored for the task of Intervention Target Estimation (ITE) is novel, to the best of my knowledge. The authors propose to repurpose some of the VGAE framework for GNN-DAG for the purpose of learning the intervention target, so the approach is not entirely novel, but their application of it to the ITE problem definitely is.

Quality
The methodology is sound and well explained. As far as I can tell, the experiments are well-designed. The method is validated against a variety of datasets and baseline methods. The empirical results convincingly support the superiority of DeepITE over existing models, particularly in terms of scalability and efficiency in handling large and complex graphs.

Clarity
The paper is well-written.

Significance
This work is significant, it may be important for root-cause analysis in systems where the causal graph is known (which is not often the case).

**Weaknesses:**

1. The authors should make it explicit from the introduction that their framework assumes that the graph is known (but not the full SCM). As far as I understand, this is not the case in Yang et al. [7]. I believe the manuscript would be clearer if it had a table summarizing the different efforts for RCA and ITE, explaining for each paper what they assume (wrt knowledge of causal model, nature of intervention, max number of nodes targeted, and any other relevant assumptions), so that a reader may situate the work more carefully.
2. The claim that the model can handle soft interventions seems uninstantiated. I quote "A probability of gamma_i = 0 being one indicates a hard intervention, whereas any other value suggests a soft intervention". Given that there are many different layers of complexity at play here, I do not believe this is a reasonable claim. For example, this is a clear instance of model misspecification, in which I do not expect there there would be consistency. For certain type of soft interventions, it is plausible that the proposed model (based on hard interventions) would erroneously predict the set of intervened nodes. Additionally, the model's inference strategy uses variational approximations, and differentiable relaxation of discrete distributions, which makes it very hard to assess whether "probability of gamma_i = 0 [equals] one".
3. I wish to have seen some form of consistency theory for the proposed method. So far it has little to no theoretical backing, even though I understand that the authors have explained well why the generative model has a nice causal semantic.
4. A main advantage of the method is to amortize across different settings, but I am not sure this claim is perfectly justified. For example, DeepITE mix performs worse than the sep variant and it performs worse than CI-RCA in table 1 (unlike written in the text!). Additionally, the authors didn't quite explain how many graphs were put together, and I didn't see in the main text the existence of an ablation study to show that with more graphs, the method performs better. I wish to have seen a assessment of the amortization to justify that it actually works.
5. It would be great if the authors could explain when they expect the model to perform well, or not, and which components are helpful for its empirical success.

**Questions:**

- Can you clarify whether the framework assumes the causal graph is known a priori, as this seems to differ from approaches like Yang et al. [7]? Would it be possible to include a table summarizing key assumptions of different RCA and ITE efforts for clearer comparative context?
- Can you substantiate the claim that the model can handle soft interventions, given the complexities associated with different types of interventions and the potential for model misspecification?
- Is there any consistency theory or additional theoretical backing for the proposed method, especially regarding its generative model and causal semantics?
- How do you justify the claim about the model's ability to amortize across different settings, particularly given the mixed performance of DeepITE in various configurations as noted in your results?
- Could you discuss under what conditions the model is expected to perform well, which components significantly contribute to its empirical success, and any limitations in its current design?

---

> ### Author Rebuttal · Authors · 2024-08-07
>
> _Q1 summarizing table_
>
> We agree that LIT does not require a given graph and will mention DeepITE requires a given graph explicitly in the introduction. In response to your suggestion, **we have provided a comparative analysis of the assumptions inherent in various methods in Table R1 of the PDF attached to our global response.** This table highlights that the primary advantages of DeepITE are its scalability and the flexible amortization during inference, allowing for adaptation without the need for retraining on new graphs and intervention targets. It also has limitations: it struggles to address confounders, and requires the graph structure as an input, despite its capacity to learn structural equations. **These two limitations have been mentioned in Appendix H (Page 20).**
>
> _Q2 soft interventions_
>
> Thank you for your valuable feedback! We have examined DeepITE's performance with various proportions of hard and soft interventions. Soft interventions are modeled by replacing the linear structural equations related to intervention targets with quadratic ones. The results, presented in Figure R1(c) of the attached PDF, show that **DeepITE is robust to different mixtures of hard and soft interventions, demonstrating its capability to handle both types effectively.** In addition, as detailed in Appendix G.3 on Page 18, we conduct further experiments based on classical ITE settings that also involve soft interventions. We will mention "soft interventions" explicitly in that section. The results, showcased in Table 5 in our paper, reinforce **the superiority of DeepITE over other baseline models in these scenarios, further confirming its effectiveness in dealing with soft interventions.**
>
> _Q3 consistency_
>
> In our paper, we primarily demonstrate that the decoder, as defined in Eq (5), can address interventional queries by modifying the adjacency matrix $A$ through graph surgery (see Propositions 1-3), since **this mechanism lays the foundation for deriving DeepITE for ITE**. Whether the estimated set of intervention targets is consistent with the ground truth relies on the properties of the amortized variational inference framework we use.
>
> On the other hand, we notice that **the consistency of VGAE for causal inference (the inverse process of ITE, see Lines 158-162) has been proven in [R4]**. Due to the time limit, we are unable to prove the consistency of DeepITE at this moment, but **will highlight this limitation in Appendix H (Page 21), identifying it as a promising area for future research:**
> > Finally, proving the consistency and identifiability of DeepITE, and more broadly in the application of VGAEs for ITE, remains an interesting avenue for future work. Notably, such theoretical guarantees have been established for VGAEs in the context of causal inference (both observational and interventional) in [R4].
>
>
> _Q4 amortization_
>
> Thank for your constructive suggestion! We have investigated the performance of DeepITE as more graphs with different sizes are trained together. Specifically, we begin with 100 graphs, each containing 50 nodes and 1000 samples, and assess the model's performance on testing graphs containing 50 nodes. We subsequently introduce additional groups of graphs: 50 graphs with 100 nodes, followed by another 50 graphs also with 100 nodes, and finally 50 graphs with 500 nodes, followed by another 50 graphs also with 500 nodes.
>
> As detailed in Figure R1(e), our findings indicate a gradual degradation in the performance of DeepITE (mix) as we incorporate more graphs of varying sizes, attributed to the amortization error. However, **this reduction in performance is minimal.**
>
> Moreover, the results in Table 1 of our original paper show that while DeepITE (mix) performs slightly worse than DeepITE (sep), it even outperforms DeepITE (sep) training exclusively on 100-node graphs in terms of Recall@1. Based on this evidence, **we maintain that the amortization process across graphs does not significantly hinder the performance of DeepITE.**
>
> _Q5 performance boundary & components contribution_
>
> Thank you for your constructive feedback! We have conducted additional ablation studies to investigate the performance of DeepITE under various conditions, specifically focusing on graph size, the number of interventions, and sample size. The findings are illustrated in Figure R1 in the attached PDF. Our results indicate that the **performance of DeepITE gradually declines as graph size and the number of interventions increase, and as sample size decreases, which aligns with our expectations.**
>
> To evaluate the contributions of different components within DeepITE, we performed a detailed ablation study, the results of which can be found in Appendix G.6 (Page 20). Our analysis revealed that the **specific designs of both the encoder and decoder play critical roles in the model's empirical success.**
>
> Lastly, **we have outlined the limitations of DeepITE in Appendix H (Page 20).**
>
> [R4] Zečević et al. Relating GNN to structural causal models, 2021.

---

> ### Author Response · Authors · 2024-08-12
> **A follow-up message about the rebuttal**
>
> Dear Reviewer HUbk,
>
> We wanted to kindly check in on the status of the rebuttal, as there are 2 days remaining for the rebuttal period. Please let us know if there is anything else we can provide to contribute to our work. We would greatly appreciate it if you could provide your insights at your earliest convenience.
>
> Thank you for your time and consideration.
>
> Best regards,
>
> Authors of DeepITE

---

> > ### Comment · Reviewer_HUbk · 2024-08-13
> >
> > I would like to thank the authors for their detailed response, and the additional experiments. I do not have additional questions, and will consider changing my score while discussing with other reviewers.

---

> > > ### Author Response · Authors · 2024-08-13
> > > **Reply to Reviewer HUbk**
> > >
> > > Thank you very much for your positive feedback. We greatly appreciate your time and thoughtful review.

---

### Official Review · Reviewer_bhQC · 2024-07-04

**Soundness:** 2
**Presentation:** 2
**Contribution:** 2
**Rating:** 4
**Confidence:** 3

**Summary:**

The paper presents DeepITE, a novel deep learning framework designed for Intervention Target Estimation (ITE) in complex systems. DeepITE addresses these issues by employing a variational graph autoencoder (VGAE) that can learn from both unlabeled and labeled data across various intervention targets and causal graphs. The framework is capable of self-supervised and semi-supervised learning, allowing it to identify intervention targets without the need for retraining with new instances. It demonstrates improved performance in Recall@k metrics and faster inference times, especially for large graphs.

**Strengths:**

1. **Originality**: DeepITE addresses a novel problem formulation in the context of causal analysis, focusing on estimating intervention targets in a manner that is not addressed by existing methods. Additionally, this method overcomes limitations of prior work by enabling collaborative learning across different instances and effectively incorporating labeled data, which was underutilized in previous approaches.
2. **Empirical Validation**: Extensive experiments are conducted, comparing DeepITE against 13 baseline methods on both synthetic datasets and real-world dataset, demonstrating its superiority in Recall@k metrics and inference time, particularly for large graphs.
3. **Presentation**: This paper clearly introduces the ITE problem and existing solutions, and elucidates the relationship between the proposed method and existing works (DAG-GNN and VACA), emphasizing the flexibility and effectiveness of DeepITE in handling graphs of different sizes and structures.

**Weaknesses:**

While the paper presents a significant contribution with the introduction of DeepITE, there are several areas where the work could be improved towards its stated goals:
1. **Theoretical Depth**: The paper could benefit from a more thorough theoretical analysis, particularly in understanding the limitations of the VGAE approach compared to traditional causal inference methods. For instance, a deeper discussion on the assumptions made by DeepITE and how they compare to assumptions in established causal frameworks would be valuable.
2. **Scalability Analysis**: While the paper mentions the efficiency of DeepITE, a more detailed analysis of its scalability with respect to the size of the graph and the number of interventions would be beneficial. This could include profiling the computational complexity and memory usage for very large graphs.
3. **Intervention Types**: DeepITE is designed to handle interventions, but the paper could provide more details on the types of interventions it can effectively estimate. For example, it would be useful to know how the model performs with different mixes of hard and soft interventions.
4. **Confounding Factors**: The paper assumes the absence of confounders, which may not hold in real-world scenarios. Future work could explore extensions of DeepITE to handle potential confounding, possibly through the integration of domain knowledge or advanced causal inference techniques.

**Questions:**

1. **Identifiability**: Does DeepITE have decidability, and can it guarantee that the optimal set of intervention variables can always be recovered from the data?
2. **Sample size**: Since this method is based on deep networks, are there any specific requirements for the amount of data samples?
3. **Robustness to random initialization**: according to algorithm 1,  this method need to be initialized by some random parameters. Does it robustness to the random initialization?

---

> ### Author Rebuttal · Authors · 2024-08-07
>
> _Q1 theory_
>
> We acknowledge the importance of theoretical analysis. However, due to the time limit, **we instead provide a comparative analysis of the assumptions inherent in various methods in Table R1 of the PDF attached to our global response.** This table highlights that the primary advantages of DeepITE are its scalability and the flexible amortization during inference, without the need for retraining on new graphs and intervention targets (ITs). However, it struggles to address confounders, and it requires the graph structure as an input, despite its capacity to learn structural equations. **These two limitations have been mentioned in Appendix H (Page 20).**
>
> _Q2 scalability_
>
> Thank you for your insightful feedback! We have provided the performance of DeepITE as a function of the graph size and the number of interventions respectively in Figs R1(a-b) in the attached PDF. Our findings indicate that while performance in terms of Recall@1 declines as the graph size increases, Recall@5 remains stable (above 0.9), suggesting that the **true ITs consistently appear among the top 5 candidates, even for graphs with 1000 nodes—a size that is already considered quite large for causal analysis**. Moreover, as the number of interventions rises, DeepITE's performance decreases gradually; however, **even with 5 interventions, the Recall@k remains at 0.825**.
>
> _Q3 intervention types_
>
> Thanks for your constructive suggestion! We have examined DeepITE's performance with various proportions of hard and soft interventions. Soft interventions are modeled by replacing the linear structural equations related to intervention targets with quadratic ones. The results, presented in Figure R1(c) of the attached PDF, show that **DeepITE is robust to different mixtures of hard and soft interventions, demonstrating its capability to handle both types effectively.** In addition, as detailed in Appendix G.3 (Page 18), we conducted experiments based on classical ITE settings with soft interventions. We will mention "soft interventions" explicitly in that section. The results, shown in Table 5 in our paper, reinforce **the superiority of DeepITE over other baseline models in dealing with soft interventions.**
>
> _Q4 confounders_
>
> **We acknowledge the importance of addressing confounders as a critical future work in Appendix H (Page 21)**. In line with other ITE works, where algorithms are initially developed for scenarios without confounders [3] before being extended to handle confounders [6], we also plan to extend DeepITE for confounders in future work.
>
> _Q5 identifiablity_
>
> In our paper, we primarily demonstrate that the decoder, as defined in Eq. (5), can address interventional queries by modifying the adjacency matrix $A$ through graph surgery (see Propositions 1-3), since **this mechanism lays the foundation for deriving DeepITE for ITE**. Whether the optimal set of ITs can be detected relies on the properties of the amortized variational inference framework we use.
>
> On the other hand, we notice that **the identifiability of VGAE for causal inference (the inverse process of ITE, see Lines 158-162) has been proven in [R4]**. Due to the time limit, we are unable to prove the identifiability of DeepITE for now, but **will highlight this limitation in Appendix H, identifying it as a promising area for future research:**
> > Finally, proving the consistency and identifiability of DeepITE, and more broadly in the application of VGAEs for ITE, remains an interesting avenue for future work. Notably, such theoretical guarantees have been established for VGAEs in the context of causal inference (both observational and interventional) in [R4].
>
>
> _Q6 sample size_
>
> Thanks for pointing it out! We have depicted the performance of DeepITE as a function of the sample size in Figure R1(d) in the attached PDF. Here we choose 10 graphs for training and change the sample size from 25 to 1000 for each graph. Our findings indicate that **DeepITE is generally robust to variations in sample size, though a larger sample size can enhance its performance**. In particular, on graphs with 50 nodes, DeepITE achieves a Recall@1 of 0.812 and a Recall@5 of 0.984 even with just 50 samples for each graph (500 samples in total). This success may stem from the collaborative learning approach in DeepITE and the relatively few parameters in the GNN-based encoder and decoder. In contrast, traditional ITE methods [3,6,10] typically require thousands of samples for a single graph and intervention set to perform well. However, it is also important to note that **the performance of DeepITE declines rapidly when the sample size is extremely small (e.g., 25 samples per graph), which aligns with our expectations**.
>
> _Q7 random initialization_
>
> DeepITE is robust to random initialization. **Since each node in a graph is initialized randomly and all graphs are trained collaboratively**, the method can converge to optimal solutions regardless of the initial parameters. **We will mention this point in our revised paper.**
>
> [R4] Zečević et al. Relating GNN to structural causal models, 2021.

---

> ### Author Response · Authors · 2024-08-12
> **A follow-up message about the rebuttal**
>
> Dear Reviewer bhQC,
>
> We wanted to kindly check in on the status of the rebuttal, as there are 2 days remaining for the rebuttal period. Please let us know if there is anything else we can provide to contribute to our work. We would greatly appreciate it if you could provide your insights at your earliest convenience.
>
> Thank you for your time and consideration.
>
> Best regards,
>
> Authors of DeepITE

---

### Official Review · Reviewer_GccQ · 2024-07-10

**Soundness:** 3
**Presentation:** 4
**Contribution:** 3
**Rating:** 6
**Confidence:** 3

**Summary:**

This paper proposes a deep learning approach for Intervention Target Estimation (ITE) which is an important problem in causal discovery and inference. The authors argue that traditional methods in this area can only independently process each instance and is computationally inefficient. To address these limitations, the authors propose a variational graph auto-encoder whose key components involve an encoder and a decoder that are specially designed to accommodate the properties of causal factorization/intervention. The authors also discuss how to instantiate the encoder and decoder with modern GNNs and use the variational inference for deriving the objective. Experiments on synthetic datasets and two real-world datasets verify the effectiveness of the model against several state-of-the-art baselines.

**Strengths:**

The paper studies an important problem whose significance is embodied by extensive applications in various domains.

The proposed method seems interesting and technically sound. The proposed components are properly justified with theoretical analysis and explanations against potential alternatives. Also, comparison with related models is sufficiently conducted.

The experiments are solid and comprehensive. The improvements achieved over the state-of-the-arts look promising.

**Weaknesses:**

The model section is not entirely clear, and some parts need further explanation.

The technical novelty is somehow weakened given that the direct usage of DAG-GNN in Sec 5.1.

The experiments lack qualitative analysis or case study to verify the model.

The presentation can be improved, especially the descriptions about prior art and the motivation in Sec. 1 (the third and fourth para are too long and involved).

**Questions:**

1. What precisely the semi-supervised learning is conducted using the model? How the labeled data is incorporated into the ELBO?

2. Can the authors provide more explanations why Recall is used as the only metric for evaluation?

3. Given the studied problem, maybe more qualitative analysis or case study are expected to further verify the model.

Minor:

1. The authors argue that GAT enables inductiveness and consider this as the advantage of GAT over other GNNs. This arguement seems problematic, since in the context of the proposed framework, the GAT can be in principle replaced by other off-the-shelf GNNs, such as basically GCN and SGC. Maybe more explanations or ablation studies are needed for jusitification on this design choice.

2. Some typos for reference:

line 106: "but this too suffers from scalability issues with large graphs"

line 109: "zeroes in on ITE"

**Limitations:**

The limitations are properly discussed in the appendix.

---

> ### Author Rebuttal · Authors · 2024-08-07
>
> Thank you very much for your positive evaluation and encouraging feedback on our paper. We deeply appreciate your constructive comments and valuable suggestions.
>
> _Q1 direct use of DAG-GNN_
>
> DeepITE does not directly use DAG-GNN. **We have explcitly discussed the relation between DeepITE and DAG-GNN on Lines 260-270 on Pages 6-7.** In summary, the key distinction lies in their objectives: DAG-GNN is designed for causal discovery (learning the DAG structure), whereas DeepITE focuses on ITE. Although both models share some similarities as variants of VGAEs, their inference models and latent spaces differ significantly. Furthermore, **our ablation study in Appendix G.6 demonstrates the superiority of DeepITE over DAG-GNN for ITE tasks.**
>
> _Q2 case study_
>
> Thanks for pointing it out! We have included a case study as follows:
> > We first present the causal graph in Figure R2. Recall that we focus on the RCA problem and we aim to identify and present the root causes of the system to users. Note that while we have labels for the root causes in our testing data, we only possess observations for the observable variables represented in the graph. Consequently, all methodologies employed can only localize observable variables as intervention targets (ITs), rather than directly identifying the root causes. For instance, RootCause 2 (a weak signal in marginal areas) can influence feature19, feature X, and feature Y. However, RootCause 3 also affects feature X. As a result, when a given method identifies either feature X as an IT, it becomes challenging to ascertain whether RootCause 2 is indeed the true root cause.
>
> > Interestingly, **DeepITE model demonstrates superior performance on this dataset by effectively identifying features exclusive to a specific root cause.** We evaluated the performance of all methods on samples 1758, 1760, and 1093, and summarized the results in Table R2. For sample 1758, where the true root cause is RootCause 2, DeepITE identifies feature19 as the IT, thus it is evident that RootCause2 should be the root cause. In contrast, other methods identify either feature X or feature Y, making it difficult to pinpoint the true root cause. Similarly, for sample 1760, DeepITE identifies feature60 as the IT, which is also exclusive to the true root cause, RootCause 3. On the other hand, in the case of sample 1093, DeepITE selects both feature60 and feature19, enabling us to conclude that both RootCause 2 and RootCause 3 are relevant root causes, a finding that aligns with the ground truth.
>
> _Q3 presentation_
>
> To address your concern, we will break up the third and fourth paragraphs into two shorter paragraphs each (e.g., the third paragraph can be divided at Line 46). Additionally, we will provide an introduction to less well-known priors, such as Jeffrey's prior, in the appendix, which will further enrich the reader's understanding of the priors.**
>
> _Q4 semi-supervised learning & labeled data_
>
> As detailed in Lines 283-309 in Section 5.3 (Page 7), the labeled data can be utilized to train the inference network, enabling it to more accurately identify intervention targets. In particular, the term $q(\gamma_i|\mathbf{x}, A)$ is replaced by the ground truth $\gamma_i$ when computing the ELBO (Eq. (14)), and an additional term is introduced to maximize the log-likelihood $\log q(\gamma_i^*|\mathbf{x}, A)$.
>
> _Q5 why use Recall_
>
> As defined in 326-328 on Page 8, Recall@k measures th proportion of true intervention targets (ITs) that are successfully captured within the top k ranked candidates proposed by each method. When k = 1, our goal is to pinpoint the intervention targets based on the highest-ranked candidate. We prioritize Recall@k because, in practice, false positives can be eliminated through further analysis, while false negatives are irrecoverable as they get lost among the numerous true negatives. This metric is widely adopted in the literature [4, 9, 29]. **We will clarify this point in the revised paper.**
>
> Furthermore, **we have incorporated additional metrics in our evaluations**, such as Root.Acc and Score in the ICASSP experiments (see Table 7), and MMD/MSE in the ablation study (see Table 8).
>
> _Q6 why use GAT_
>
> We acknowledge that any inductive spatial GNN can be used as the inference network in DeepITE, and **will mention this point in our paper**. However, **we choose GAT since it is more flexible compared to GCN and SGC** [R3]. This flexibility stems from GAT's ability to dynamically weigh the importance of different nodes, thus allowing the variational distribution given by the inference network better approximate the exact posterior distribution. **This advantage has been demonstrated in Appendix G.6 (Page 20), where we replace the GAT encoder with the encoder of DAG-GNN, a type of GCN [15], and show the benefits of using GAT.**
>
> _Q7 typos_
>
> We will fix the typos accordingly.
>
> [R3] Veličković, et al, Graph Attention Networks, ICLR 2018.

---

> ### Author Response · Authors · 2024-08-12
> **A follow-up message about the rebuttal**
>
> Dear Reviewer GccQ,
>
> We wanted to kindly check in on the status of the rebuttal, as there are 2 days remaining for the rebuttal period. Please let us know if there is anything else we can provide to contribute to our work. We would greatly appreciate it if you could provide your insights at your earliest convenience.
>
> Thank you for your time and consideration.
>
> Best regards,
>
> Authors of DeepITE

---

> > ### Comment · Reviewer_GccQ · 2024-08-13
> >
> > Thanks for the discussions. Please incorporate the case study results and clarification into the paper.
> >
> > For the different from DAG-GNN, the main theoretical results of this paper are from DAG-GNN. While this work applies the method to different problem settings, the technical novelty and contributions would be limited in this sense.
> >
> > For Recall metric, the authors use different metrics for different datasets without enough justification. I understand what Recall means, but the rebuttal fails to properly justify why only Recall is used for Protein/Synthetic and the other metrics are used for ICASSP. This questions the robustness of the results.
> >
> > The argument that GAT is inductive is not rigorously correct, since common GNNs (e.g., GCN and SGC) are all applicable for inductive learning. The inductiveness is not a unique advantage of GAT.

---

> ### Author Response · Authors · 2024-08-13
> **Reply to Reviewer GccQ**
>
> We greatly appreciate your valuable suggestions and the time you've taken to provide detailed feedback. We will carefully incorporate the case study results and clarifications into our paper.
>
> > For the different from DAG-GNN, the main theoretical results of this paper are from DAG-GNN. While this work applies the method to different problem settings, the technical novelty and contributions would be limited in this sense.
>
> We would like to clarify that DAG-GNN **does not prove any of the theorems presented in our paper**, even within their problem settings. Instead, DAG-GNN provides two theorems specifically related to their proposed acyclicity constraints.
>
> > For Recall metric, the authors use different metrics for different datasets without enough justification. I understand what Recall means, but the rebuttal fails to properly justify why only Recall is used for Protein/Synthetic and the other metrics are used for ICASSP. This questions the robustness of the results.
>
> Thank you for raising this concern. The additional metrics used for the ICASSP dataset were selected due to the unique characteristics and requirements of this particular dataset. As a result, these additional metrics **could not be applied to to the Protein/Synthetic datasets** in our study.
>
> As explained in our rebuttal, we prioritize Recall@k because, in practice, false positives can be eliminated through further analysis, while **false negatives are irrecoverable** as they get lost among the numerous true negatives. This approach ensures that critical detections are not overlooked, which is why Recall@k is emphasized for certain datasets in our study.
>
> On the other hand, it’s important to note that Recall@k **inherently includes a ranking among all candidate intervention targets** (ITs). A higher Recall@1 indicates that the true IT is ranked first, which implies that, with an appropriate threshold, the precision will also be very high. This is another reason why we focused on Recall@k in our analysis.
>
> > The argument that GAT is inductive is not rigorously correct, since common GNNs (e.g., GCN and SGC) are all applicable for inductive learning. The inductiveness is not a unique advantage of GAT.
>
> Thank you for pointing this out. We agree that GCN and SGC are also inductive, as they, along with GAT, fall under the category of spatial GNNs rather than spectral GNNs, making them all applicable for inductive learning.
>
> As mentioned in our rebuttal, we **choose GAT** not because only GAT is inductive, but **because it is more flexible compared to GCN and SGC** [R3]. This flexibility stems from GAT's ability to dynamically weigh the importance of different nodes, thus allowing the variational distribution given by the inference network better approximate the exact posterior distribution. This point is also validated in our experiments in Appendix A.6.
>
> Once again, we sincerely thank you for your thoughtful feedback.

---

### Official Review · Reviewer_q8ww · 2024-07-15

**Soundness:** 3
**Presentation:** 2
**Contribution:** 2
**Rating:** 5
**Confidence:** 4

**Summary:**

Given a causal graph, this paper describes an autoencoder-based approach specifically designed for the purpose of designing intervention targets. This algorithm aims to be data and computation-efficient by by-passing the task of having to recover the causal graph, and also by incorporating specific architectural designs for the auto-encoder.

**Strengths:**

The main strengths of this paper are as follows:

1. Empirical results: the results of this paper seem to be very convincing, and there is comparison to many related baselines.

2. The method is concise and easy to implement/understand and relate to prior work.

**Weaknesses:**

The weaknesses of this paper are as follows:

1. The code is not available -- the main contributions of this work are experiments and empirical evaluation, which are quite strong. In order to truly assess this work, code availability, while not the only factor, is very important.

2. Lack of clarity: a) There is lack of clarity in terms of the datasets and evaluations (see questions below); b) there is lack of clarity on the competing methods and how this method fits within the general framework. For example, it would be great if RCA, and the main ITE baselines can be formally described in mathematical terms through a simple example (either in the main text or the appendix).

3. Evaluation: the authors seem to not discuss a very trivial and intuitive baseline regarding prediction methods with sparsity such as Lasso (please see questions below).

**Questions:**

I have a few questions for the authors:

1. What is the purpose of the noise variable \zeta, and why is it modeled as a separate variable from the \mathbf{u} variable? From my experience, the exogenous variables are simply independent noise terms that are provided as input to the functional causal model.

2. How do the authors know the true intervention targets of the protein dataset?

3. Have the authors considered the following baseline: given M interventional datasets and an observational dataset, label each point with label G corresponding to which interventional dataset it comes from. Then, train a sparse classification (feature selection) method such as sparse logistic regression to predict G from the features X when combining an interventional and an observational dataset. Then, the selected features will be the Markov Blanket of G. I was wondering what the author's thoughts are on this simple baseline.

4. In line 300, the authors mention: "Once trained, the inference model of DeepITE becomes equipped to evaluate individual new samples
against different causal graphs, directly deducing the intervention targets and thus circumventing
the necessity of retraining for each new scenario...."; could the authors elaborate on this point? My understanding is that the model assumes a given causal graph.

**Limitations:**

From what I see, there is no discussion on this model's limitation in the text.

---

> ### Author Rebuttal · Authors · 2024-08-07
>
> _Q1 code availability_
>
> We are committed to open-sourcing the code upon acceptance of the paper.
>
> _Q2 lack of clarity_
>
> 1. Datasets and Evaluations: We have provided descriptions of the datasets used, including their sources, preprocessing steps, and evaluation metrics in Appendix G (please refer to Appendix G.2-G.5)
> 2. Competing Methods: In Appendix B (pages 13–14), we have provided a review of the existing methods in XAI and RCA. **This section includes a concise explanation of how each method operates**. While we acknowledge that mathematical formulations can be rigorous, we believe that explaining these methods in plain language enhances clarity and comprehension. Additionally, **we have referred readers to Appendix B within the experiment section (see Line 316 on Page 7) for easy access.**
>
> _Q3 observation noise_
>
> DeepITE includes the observation noise variable $\epsilon$ that reflects uncertainty not present in the true SCM. In the true SCM, observed variables are deterministic functions of their exogenous variables and parent nodes via the structural equations (SEs). Since DeepITE does not have access to the true SEs or the distribution of the exogenous variables, it assumes the SEs can be approximated as $Dec(\mathbf{u}, \gamma, A) = f_2((I- A_\mathcal{I}^T)^{-1}f_1(\mathbf{u}))$ (Eq. (7)), with learnable $f_1$ and $f_2$. Thus, $\epsilon$ represents the uncertainty in the estimated observational distribution resulting from this approximation. Note that $\epsilon$ has also been used in VACA [21] for the same sake. **We have clarified this point on Lines 222-224 in Section 5.1.**
>
> _Q4 protein dataset_
>
> We clarify that the true intervention targets of the protein dataset are known because we utilize the dataset from IGSP [R1], which provides this information in Appendix E in their paper. Specifically, this datasets contain measurements of proteins and phospolipids under different interventional environments. In each environment, signaling nodes are inhibited or activated. Hence, **these sites form intervention targets.** This dataset has been previously employed in [3,6,10] for ITE. We will explicitly mention this in the dataset introduction in Appendix G.4.
>
> _Q5 Lasso_
>
> We respectfully believe that Lasso does not align with the scope of ITE as outlined Definition 1 on Page 4. First, **Lasso cannot incorporate the causal graph structure available in ITE**. The prediction G can be an arbitrary variable in the graph and there is no clear guideline on how to choose G given the observed data. Moreover, **Lasso is designed to identify correlations rather than causal relationships, and so it is typically used for undirected graphical models [R2]**. Consequently, Lasso may select all variables correlated with G, rather than correctly identifying the intervention targets that directly impact G.
>
> Actually, Lasso has been used for RCA in [8], but only for **bipartite** directed graphs where causal and effect relationships between features and predictions have been provided. In contrast, our approach is designed to handle more complex causal graph structures where the predictions are unknown.
>
> _Q6 given causal graph_
>
> You are correct that DeepITE assumes a given causal graph (see Defintion 1 on Page 4). However, once trained, it can evaluate new samples on new causal graphs without needing to retrain. By directly inputting the sample and the new graph structure into the inference network, we can derive the intervention targets. This stands in contrast to classical ITE methods, which require retraining for each new causal graph and each new set of intervention targets, even if the graph remains unchanged.
>
> _Q7 limitations_
>
> We have included a discussion of the limitations in Appendix H (Page 21).
>
> [R1] Wang et al, Permutation-based Causal Inference Algorithms with Interventions, NIPS 2017.
>
> [R2] Meinshausen et al, High-dimensional Graphs and Variable Selection with the Lasso, Ann. Statist., 2006.

---

> > ### Comment · Reviewer_q8ww · 2024-08-13
> >
> > I thank the reviewers for detailed responses to my questions, and for addressing some of my concerns.
> >
> > Regarding Q5:
> > - I think we may have a misunderstanding. If G is an auxiliary discrete variable (with discrete values denoting the interventional environment a data point came from), which has the same value for each point in the same interventional environment, a method like Lasso would indeed find all variables that are associated with G (it will yield the variables in the Markov Blanket of G, we can term them MB(G)). One can also assume that G is always a cause (by definition it cannot be an effect of any variable). Then, MB(G) will be either the direct effects of G, or parents of the direct effects. Then, one can think about using the known causal graph to distinguish these sets of variables.
> >
> > Regarding Q6:
> > - Can the authors briefly elaborate why it is possible to generalize to new, previously-unseen, causal graphs when using their framework?
> >
> > Thank you very much.

---

> ### Author Response · Authors · 2024-08-12
> **A follow-up message about the rebuttal**
>
> Dear Reviewer q8ww,
>
> We wanted to kindly check in on the status of the rebuttal, as there are 2 days remaining for the rebuttal period. Please let us know if there is anything else we can provide to contribute to our work. We would greatly appreciate it if you could provide your insights at your earliest convenience.
>
> Thank you for your time and consideration.
>
> Best regards,
>
> Authors of DeepITE

---

> ### Author Response · Authors · 2024-08-13
> **Reply to Reviewer q8ww**
>
> Thank you very much for your valuable feedback.
>
> >Regarding Q5:
>
> >- I think we may have a misunderstanding. If G is an auxiliary discrete variable (with discrete values denoting the interventional environment a data point came from), which has the same value for each point in the same interventional environment, a method like Lasso would indeed find all variables that are associated with G (it will yield the variables in the Markov Blanket of G, we can term them MB(G)). One can also assume that G is always a cause (by definition it cannot be an effect of any variable). Then, MB(G) will be either the direct effects of G, or parents of the direct effects. Then, one can think about using the known causal graph to distinguish these sets of variables.
>
> Thank you very much for your clarification. We now understand that this is quite similar to the method implemented in [3], and we have compared DeepITE with it in our analysis.
>
> >Regarding Q6:
>
> >- Can the authors briefly elaborate why it is possible to generalize to new, previously-unseen, causal graphs when using their framework?
>
> Sure. In simple words, after training the VGAE collaboratively using graphs with different sets of intervention targets, different structures, and even different sizes, the inference model (i.e., the encoder) in the VGAE successfully learns a mapping whose input is the observed data $\mathbf{x}$ and the adjacency matrix $A$ and output is the distribution of intervention indicator $\gamma$. As a result, when we replace the adjacency matrix $A$ by that of an unseen graph, the encoder can still output the distribution of the intervention indicator.
>
> Thank you again for your thoughtful response, as they have provided valuable insights for improving our paper.

---

### Author Rebuttal · Authors · 2024-08-07

**Global response to all reviewers**:

We sincerely thank all the reviewers for their valuable suggestions. We are delighted by the unanimous recognition of our work and appreciate the reviewers' positive feedback on the carefully designed network architecture and extensive experiments in DeepITE.

We have thoroughly reviewed each of the reviewers' questions and suggestions, and we are grateful for their patience and diligence. In response, we have **conducted additional experiments** to evaluate the performance of DeepITE as a function of (a) graph sizes, (b) the number of interventions, (c) the mixture proportion of soft and hard interventions, (d) sample size for each graph, and (e) the number of mixed graphs, added new analyses to address any lingering questions, and emphasized the significance of our work. We have also **included a summary table** that highlights the characteristics of all compared methods and **a detailed case study** to further illustrate the practical application and effectiveness of our approach.

In the following rebuttal, we address each reviewer's comments individually. The reviewer's comments are presented in italics, followed by our response. Quotations from the revised paper are included in markdown quotation mode. Unless otherwise specified, all references to pages, equations, sections, and bibliographic citations relate to the revised paper. Additionally, figures, tables, and citations prefixed with "R" (e.g., [R1]) are newly added citations in this rebuttal. All newly added images and a table are enclosed within a separate single-page PDF attached to this global response. We will incorporate the suggested revisions into the final camera-ready version to enhance the clarity and persuasiveness of our paper.

Once again, we would like to express our gratitude to the reviewers for their insightful feedback, which has helped us identify areas for improvement and refine our work. We welcome any further insights or concerns that would contribute to enhancing the paper according to the reviewers' perspectives.

---

### Decision · Program_Chairs · 2024-09-25

**Decision:**

Accept (poster)

**Comment:**

I recommend the work "DeepITE" for acceptance based on its significant contributions and overall positive feedback. The work introduces a novel framework for intervention target estimation using a variational graph autoencoder (VGAE) and is empirically validated to outperform baseline methods, demonstrating superior scalability and efficiency. The paper is well-written and connects the new model to existing methods.

The score of this work has benefited a lot from the discussion between authors and reviewers. Many points have been successfully and extensively clarified and addressed by the authors.

However, there are some weaknesses that need to be addressed before the final camera-ready version. These include , clarifying assumptions related to the causal graph and intervention types, improving the theoretical analysis concerning scalability and confounder handling and providing accessible implementation code to enhance reproducibility. Addressing these points will strengthen the paper and ensure its contribution is fully validated.